# Depletable peroxidase-like activity of Fe$_3$O$_4$ nanozymes accompanied with separate migration of electrons and iron ions

Haijiao Dong [1,2], Wei Du [1,2], Jian Dong[1], Renchao Che [3,4,5], Fei Kong[1,2], Wenlong Cheng [6,7], Ming Ma [1,2] ✉, Ning Gu [1,2] ✉ & Yu Zhang [1,2] ✉

As pioneering Fe$_3$O$_4$ nanozymes, their explicit peroxidase (POD)-like catalytic mechanism remains elusive. Although many studies have proposed surface Fe$^{2+}$-induced Fenton-like reactions accounting for their POD-like activity, few have focused on the internal atomic changes and their contribution to the catalytic reaction. Here we report that Fe$^{2+}$ within Fe$_3$O$_4$ can transfer electrons to the surface via the Fe$^{2+}$-O-Fe$^{3+}$ chain, regenerating the surface Fe$^{2+}$ and enabling a sustained POD-like catalytic reaction. This process usually occurs with the outward migration of excess oxidized Fe$^{3+}$ from the lattice, which is a rate-limiting step. After prolonged catalysis, Fe$_3$O$_4$ nanozymes suffer the phase transformation to γ-Fe$_2$O$_3$ with depletable POD-like activity. This self-depleting characteristic of nanozymes with internal atoms involved in electron transfer and ion migration is well validated on lithium iron phosphate nanoparticles. We reveal a neglected issue concerning the necessity of considering both surface and internal atoms when designing, modulating, and applying nanozymes.

Since the first discovery of Fe$_3$O$_4$ nanoparticles (NPs) with intrinsic peroxidase (POD)-like activity in 2007, nanomaterial-based artificial enzymes (nanozymes) and their extensive applications have rapidly attracted attention over the past decade[1–10]. Recently, research efforts on nanozymes have gradually shifted from application-oriented to mechanism-oriented[11–15]. For example, single-atom nanozymes centered on different metal species have been synthesized with well-defined structures and coordination environments, which facilitate the identification of catalytic centers and unravel the catalytic mechanisms at the atomic level[16–19]. Besides, the high substrate selectivity of nanozymes has been achieved by the bionic principle of natural substrate channeling and stepwise screening or by molecular blotting techniques[20,21]. Given the intricate structure-activity relationships and restricted characterization techniques, however, it is still challenging to understand the explicit mechanism of most nanozymes[3,4].

Despite being a pioneering nanozyme, research on the catalytic mechanism of the POD-like activity of Fe$_3$O$_4$ NPs is still limited[10,14,15,22–25]. To date, it is generally accepted that high-reactive hydroxyl radicals (·OH) generated by Fenton-like reactions (Eqs. (1) and (2)) involving the surface Fe$^{2+}$ under acid conditions contribute to the POD-like activity of Fe$_3$O$_4$ NPs[24,25]. Although the questioning that the POD-like activity of magnetite is mediated by adventitious metal traces, it has been experimentally verified (including this paper) to exclude the interference of metal traces or the leaching effect of Fe ions in reaction solution[10,25–28]. Similar to natural horseradish peroxidase, Fe$_3$O$_4$ nanozymes follow the ping-pong mechanism and

[1]State Key Laboratory of Bioelectronics, School of Biological Science and Medical Engineering, Southeast University, Nanjing 210096, P. R. China. [2]Jiangsu Key Laboratory for Biomaterials and Devices, School of Biological Science and Medical Engineering, Southeast University, Nanjing 210096, P. R. China. [3]Laboratory of Advanced Materials, Fudan University, Shanghai 200438, P. R. China. [4]Shanghai Key Lab of Molecular Catalysis and Innovative Materials, Fudan University, Shanghai 200438, P. R. China. [5]Department of Materials Science, Fudan University, Shanghai 200438, P. R. China. [6]Department of Chemical Engineering, Faculty of Engineering, Monash University, Clayton, VIC, Australia. [7]The Melbourne Centre for Nanofabrication, Clayton, VIC, Australia. ✉e-mail: maming@seu.edu.cn; guning@seu.edu.cn; zhangyu@seu.edu.cn

Michaelis–Menten kinetics[10]. In addition, their catalytic performances are influenced by particles size, morphology, lattice structure, doping, surface modification, substrates used, and the catalytic environment exposed, all of which could affect the surface-active sites by altering the surface chemistry[10,22,23,29–32]. Other individual studies have investigated the absorption, activation, and desorption processes of substrates (e.g., $H_2O_2$ and 3, 3′, 5, 5′-tetramethylbenzidine (TMB)) on the surface of $Fe_3O_4$ at the atomic level based on density functional theory and developed some descriptors to predict their POD-like activity[14,15].

$$Fe^{2+} + H_2O_2 \rightarrow Fe^{3+} + \cdot OH + OH^{-} \qquad k_1 = 76 \ (mol/L)^{-1}s^{-1} \quad (1)$$

$$Fe^{3+} + H_2O_2 \rightarrow Fe^{2+} + \cdot OOH + H^{+} \qquad k_2 = 0.002 \ (mol/L)^{-1}s^{-1} \quad (2)$$

The above mechanistic studies share a theoretical premise: only the surface-active sites play a decisive role in the enzymatic-like property of nanozymes since catalysis occurs mainly on the particle surface or interface. This view is now widely recognized and works for most types of nanozymes[1,2,4,11,32,33]. For example, in a recent controversial question regarding how to define nanozyme concentration, Liu et al. argued that considering the whole particle or all atomic units within a particle as an enzyme unit would overestimate and underestimate the catalytic activity of nanozymes, respectively, because it is the surface atoms that are truly the catalytic active sites[33]. However, in the Fenton-like reactions triggered by $Fe_3O_4$ nanozymes, we noticed that the reaction rate constant of Eq. (1) is much higher than that of Eq. (2), which implies that the surface-active $Fe^{2+}$ is hardly recovered after being oxidized. This irreversible oxidation of surface $Fe^{2+}$ prompts us to ponder if only the surface atoms of the nanozymes, particularly for metal oxide nanozymes, act in enzymatic-like catalysis, would these active sites be exhausted after long-term catalysis, rendering the nanozymes inactive? Nevertheless, no relevant studies can conclusively answer this crucial question.

Here, we propose a neglected issue regarding the POD-like mechanism of nanozymes by characterizing the chemical composition and catalytic activity of the recycled $Fe_3O_4$ NPs participating in cyclic POD-like catalysis. Both surface and interior $Fe^{2+}$ are found to impart POD-like properties to $Fe_3O_4$ nanozymes. Generally, $Fe^{2+}$ inside the particle could transfer its electron to the surface layer, regenerating the surface $Fe^{2+}$ and sustaining the catalytic reaction. This process is usually coupled with the outward migration of excess oxidized $Fe^{3+}$, which is probably a rate-limiting step. As the catalysis continues, $Fe_3O_4$ is slowly oxidized into $\gamma$-$Fe_2O_3$ accompanying the depleted enzyme-like activity, similar to the conventional low-temperature oxidation of magnetite, only with different electron receptors. This self-depleting characteristic of nanozymes with internal atoms involved in electron transfer and ion migration is further demonstrated by a typical model material, lithium iron phosphate ($LiFePO_4$), which contains redox-active metal sites and mobile lithium ions ($Li^+$) encapsulated in a rigid phosphate network. This paper reveals that internal atoms may also contribute to nanozyme-catalyzed reactions even though these reactions occur on the surface of NPs, which is thought-provoking when designing, regulating, and applying nanozymes.

## Results and discussion
### Synthesis and characterization of IONPs
Near-spherical magnetite nanoparticles ($Fe_3O_4$ NPs) with an average diameter of $10.16 \pm 0.12$ nm (Supplementary Fig. 1a) were synthesized using the chemical coprecipitation method[24]. Maghemite ($\gamma$-$Fe_2O_3$) and hematite ($\alpha$-$Fe_2O_3$) NPs were derived by calcining the $Fe_3O_4$ NPs powder at 200 and 650 °C for 2 h, respectively (Fig. 1a)[34]. X-ray diffractometer (XRD) and Raman spectra (Supplementary Fig. 1b, c) show the successful synthesis of these three iron oxide NPs (IONPs). These IONPs were uniformly dispersed in an aqueous solution at pH of 3 by

ultrasonication (Supplementary Fig. 1d). Their POD-like activities were assessed using different colorimetric substrates, including TMB, 2, 2′-azino-bis(3-ethylbenzothiazoline-6-sulfonic acid) (ABTS), and o-Phenylenediamine (OPD), in the presence of $H_2O_2$. The results show that their catalytic activity followed the order of $Fe_3O_4$ NPs >> $\gamma$-$Fe_2O_3$ NPs > $\alpha$-$Fe_2O_3$ NPs (Supplementary Fig. 2). To better quantify their POD-like activity, we calculated their specific activity ($a_{nano}$) according to the specified method[35,36], which were 1.79, 0.45, and 0.03 U·mg$^{-1}$, respectively (Fig. 1b). As previously reported[10,24], the higher catalytic ability of $Fe_3O_4$ NPs is attributed to the ·OH arising from the surface $Fe^{2+}$-initiated Fenton-like reaction (Supplementary Figs. 3 and 4). The negligible $a_{nano}$ of $\alpha$-$Fe_2O_3$ NPs compared with $\gamma$-$Fe_2O_3$ NPs is ascribed to the structural effect of the crystal phases[37]. Briefly, $\gamma$-$Fe_2O_3$ possess cation vacancies at its octahedral positions and most of these vacancies are located on the particle surface, which can favor the adsorption of the substrate $H_2O_2$, resulting in a relatively higher POD-like activity (Supplementary Fig. 5). However, these vacancies do not exist on the surface of $\alpha$-$Fe_2O_3$ due to the change of crystal structure caused by the higher calcination temperature[37].

### Cyclic POD-like catalysis of $Fe_3O_4$ NPs
To investigate whether the surface $Fe^{2+}$ of $Fe_3O_4$ NPs is depleted after participating in prolonged catalysis, we continuously increased the amount of substrate TMB under sufficient $H_2O_2$ with as-synthesized three IONPs as continuous catalysts, and monitored the absorbance changes of TMB oxidation products at 650 nm within 12 h. From Supplementary Fig. 6d–g, even though the TMB was increased from 0.087 to 0.52 mM, the $Fe_3O_4$ NPs were still able to continuously and rapidly engage in the catalytic reaction for a long duration (≥12 h) without showing signs of depletion. We speculated two reasons: (1) the amount of substrate is still too low to completely consume the surface-active $Fe^{2+}$ and (2) the $Fe^{2+}$ within $Fe_3O_4$ NPs provides the impetus for continuous catalysis.

Cyclic POD-like catalytic assays (Fig. 1c) were carried out as validation, which could provide sufficient substrates for $Fe_3O_4$ NPs to keep exerting their POD-like capacity. We evaluated the $a_{nano}$ of the recycled $Fe_3O_4$ NPs within 5 days. The results show that the catalytic ability of $Fe_3O_4$ NPs decreased to a level comparable to that of $\gamma$-$Fe_2O_3$ NPs after five days of cyclic catalysis, while the changes of $\gamma$-$Fe_2O_3$ NPs were negligible (Fig. 1d and Supplementary Fig. 7). The impact of leached Fe ions in acidic medium on the catalytic activity of IONPs has been excluded (Supplementary Fig. 8 and Supplementary Table 1). It pushed us to wonder how the surface-active $Fe^{2+}$ of $Fe_3O_4$ NPs alone could sustain the TMB oxidation for up to 100 h. Conceivably, if only the surface-active sites are responsible for the enzyme-like performance, nanozymes will deactivate when the surface-active sites are exhausted (Supplementary Fig. 9).

To reveal the potential reasons for the sustained catalytic capacity of $Fe_3O_4$ NPs, we characterized the physicochemical properties of the recycled $Fe_3O_4$ NPs using different methodologies. The electron energy-loss spectra are a useful tool for revealing the chemical and oxidation state information of iron oxide at high spatial resolution[38]. In general, the peaks of the transition metal L-edge shift toward higher energy loss with an increasing oxidation state. For iron oxide species, the area ratios of Fe $L_3/L_2$ also increase with increasing Fe valence[39]. As shown in Fig. 1e, both of the $Fe_3O_4$ NPs before and after 5 days of cyclic POD-like reactions showed two peaks related to the Fe $L_3$ and $L_2$, with an energy gap of about 13 eV between the two white lines. However, approximately 1.4 eV chemical shift toward high energy loss could be observed from $Fe_3O_4$ NPs to recycled $Fe_3O_4$ NPs. In addition, the Fe $L_3/L_2$ area ratios also increased from 4.7 to 6.1, which indicates the increase in the Fe oxidation state of $Fe_3O_4$ NPs after 5 days of POD-like catalysis[40]. A similar finding was obtained from the XPS analysis of the recycled $Fe_3O_4$ NPs on days 0, 1, 3, and 5 of the cyclic catalysis. The X-ray penetration depth of the analyzed sample ranges from 2 to 10 nm

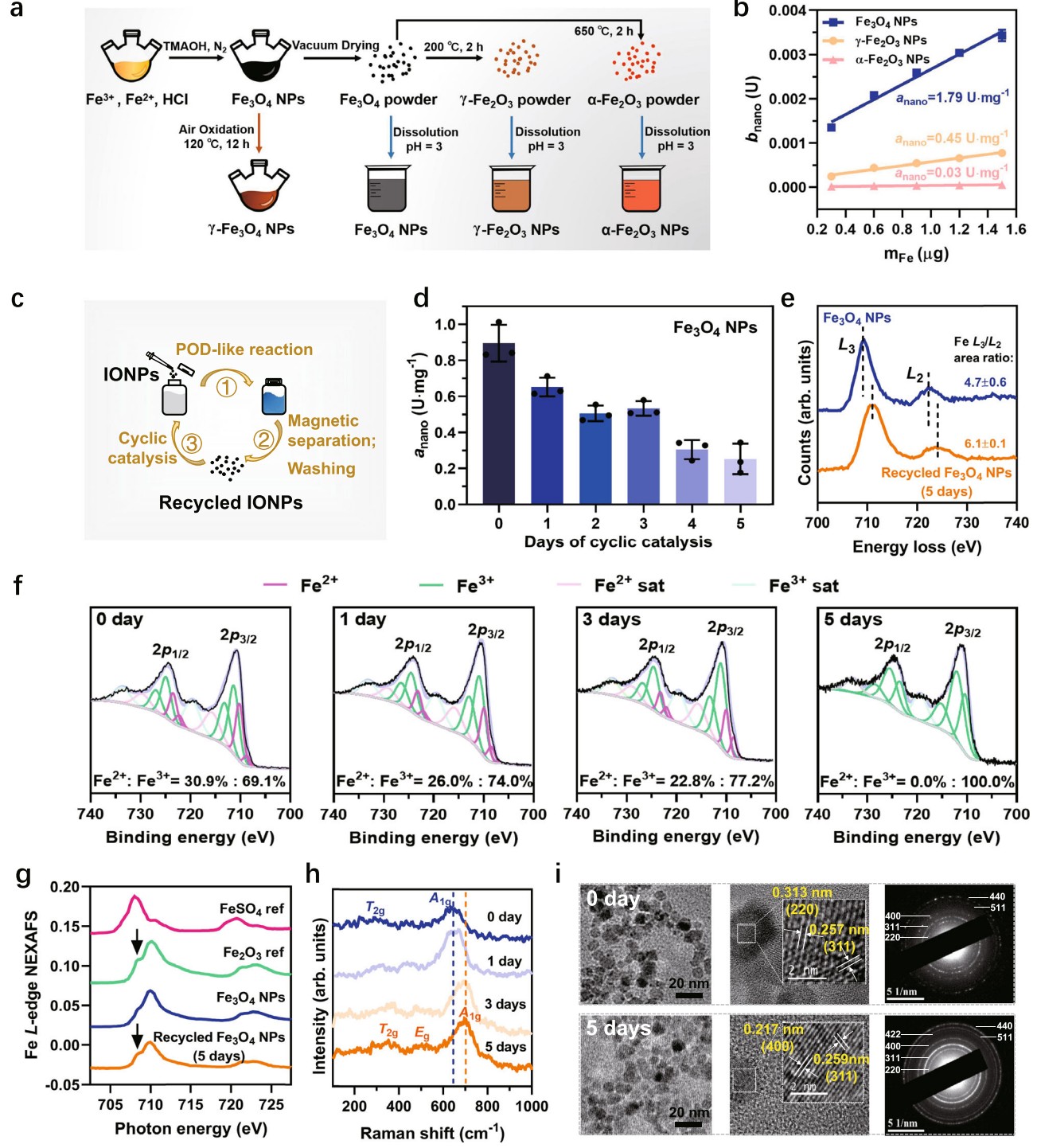

**Fig. 1 | The synthesis of IONPs and cyclic POD-like catalysis. a** Illustration of the synthesis process of IONPs. **b** The specific activity ($a_{nano}$) of these three IONPs with TMB as colorimetric substrates. **c** Diagram of the cyclic catalysis assay. **d** Kinetic study of $a_{nano}$ values of $Fe_3O_4$ NPs with the days of cyclic catalytic reaction. Error bars represent standard deviation from three independent measurements. **e** Comparison of Fe $L_{2,3}$ spectra of $Fe_3O_4$ NPs before and after 5 days of cyclic POD-like reactions. **f** The fitted Fe2p XPS spectra of $Fe_3O_4$ NPs recycled after catalysis on days 0, 1, 3, and 5. **g** The Fe L-edge NEXAFS spectra of $Fe_3O_4$ NPs and recycled $Fe_3O_4$ NPs after 5 days of catalysis in comparison with the reference spectra of $FeSO_4$ and $Fe_2O_3$. **h** Raman spectra of $Fe_3O_4$ NPs recycled after catalysis on days 0, 1, 3, and 5. **i** TEM, HRTEM images, and SAED pattern of $Fe_3O_4$ NPs and recycled $Fe_3O_4$ NPs after 5 days of catalysis. Images were collected at least three times for each type of NPs.

for the XPS technique. Since the average diameter of as-synthesized $Fe_3O_4$ NPs is around 10 nm, the Fe valence state obtained from the Fe2p fitting analysis can be approximated as the oxidation state of individual $Fe_3O_4$ NPs. As shown in Fig. 1f, the $Fe^{2+}$ in $Fe_3O_4$ NPs decreased from the original 30.9% to 0% with the extension of cyclic

catalytic days, suggesting that the interior $Fe^{2+}$ may also be oxidized to $Fe^{3+}$ in the successive POD-like reactions.

The phase transformation of $Fe_3O_4$ NPs caused by internal $Fe^{2+}$ oxidation was demonstrated by the near-edge X-ray absorption fine structure (NEXAFS) spectroscopy. Figure 1g shows the Fe L-edge

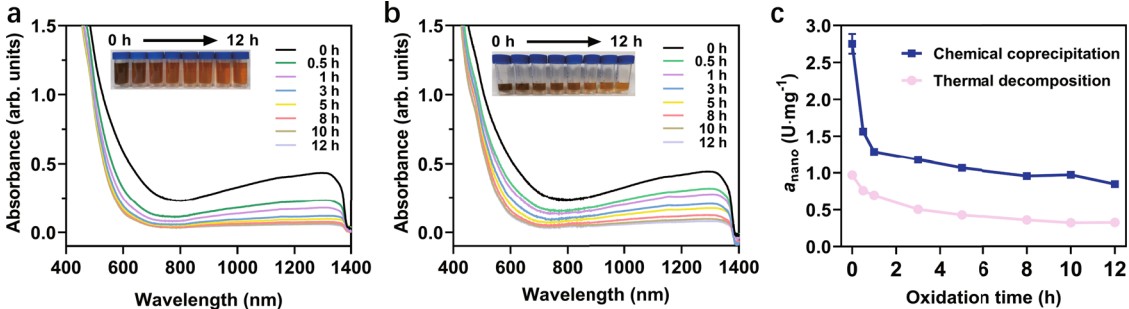

**Fig. 2 | The aeration oxidization kinetics of $Fe_3O_4$ NPs.** Variation of UV-Vis-NIR absorption of **a** cc-$Fe_3O_4$ NPs and **b** TD-$Fe_3O_4$ NPs with aeration oxidation time. Insets are photos of the suspensions corresponding to oxidation times at 0, 0.5, 1, 3, 5, 8, 10, and 12 h. All spectra and photos were obtained at the same Fe concentration. **c** Changes in $a_{nano}$ of the oxidized cc-$Fe_3O_4$ NPs and TD-$Fe_3O_4$ NPs during the aeration oxidation. Error bars represent standard deviation from three independent measurements.

NEXAFS spectra of the control $Fe_3O_4$ NPs and the recycled $Fe_3O_4$ NPs after 5 days of catalysis, in comparison with two reference spectra of $FeSO_4$ and $Fe_2O_3$. The increased splitting $L_3$ peak near 708 eV for the recycled $Fe_3O_4$ NPs coincides with reference $Fe_2O_3$[41–43]. Besides, in the Raman spectra of the recycled $Fe_3O_4$ NPs, the feature of the $A_{1g}$ mode band gradually shifted from 660 to 700 $cm^{-1}$ (Fig. 1h), corresponding to a transition from magnetite to maghemite[44,45]. In addition, transmission electron microscopy (TEM) images (Fig. 1i) and XRD pattern (Supplementary Fig. 10) show that the influence of this transformation on the particle morphology, size, and lattice structure is negligible.

Based on the above characterization results, we conclude that not only the surface $Fe^{2+}$ but also the interior $Fe^{2+}$ of the $Fe_3O_4$ nanozymes were gradually oxidized by prolonging the reaction time. Simultaneously, the catalytic activity of the recovered NPs gradually decreases with the increase of their oxidation state. Therefore, we suggest that the involvement of $Fe^{2+}$ inside the particles is responsible for the prolonged catalytic capacity of $Fe_3O_4$ nanozymes. Specifically, as shown in Supplementary Fig. 11, when the surface $Fe^{2+}$ is oxidized to $Fe^{3+}$ by the Fenton-like reaction, the adjacent $Fe^{2+}$ inside the particle will continuously transfer its electron outward via the $Fe^{2+}$-O-$Fe^{3+}$ chain in the lattice to maintain the catalytic activity of the surface Fe atoms. However, this replenishment of electrons is not infinite. When all the interior $Fe^{2+}$ are oxidized to $Fe^{3+}$, the $Fe_3O_4$ phase is transformed to γ-$Fe_2O_3$ without electrons being transferred to the surface, resulting in the reduction of catalytic activity or even inactivation.

### Aeration oxidation kinetics of $Fe_3O_4$ NPs

We assume that the oxidation of $Fe_3O_4$ nanozymes induced by POD-like catalysis is comparable to the traditional low-temperature (<200 °C) air oxidation of magnetite since the crystal structure of both remains unchanged during the oxidation process[46]. Both magnetite and maghemite contain 32 O atoms per unit cell. The difference is that the former contains 24 Fe atoms (16 $Fe^{3+}$ and 8 $Fe^{2+}$), while the latter has only 21.33 Fe atoms (all $Fe^{3+}$). Namely, once 8 $Fe^{2+}$ in magnetite are oxidized to 8 $Fe^{3+}$ releasing 8 electrons, a charge imbalance will occur (Eq. (3)). To maintain electroneutrality, 2.67 $Fe^{3+}$ have to migrate to the crystal surface, leaving the cation vacancies (Eq. (4))[46]. The outward moving $Fe^{3+}$ will coordinate with the surface absorbed $O^{2-}$ that is ionized by the electrons generated by the oxidation of $Fe^{2+}$ to $Fe^{3+}$, and form a film of the solid solution of γ-$Fe_2O_3$ in $Fe_3O_4$[34,47–49]. Therefore, the phase transformation of $Fe_3O_4$ to γ-$Fe_2O_3$ is a single-phase topological reaction accompanied by the separate migration of electrons and excess $Fe^{3+}$[34,47,49].

$$Fe_{16}(III)Fe_8(II)O_{32} \rightarrow Fe_{24}(III)O_{32}{}^{+8} + 8e^- \qquad (3)$$

$$Fe_{24}(III)O_{32}{}^{+8} \rightarrow \gamma - Fe_{21.33}(III)\square_{2.67}O_{32} + 2.67Fe(III) \qquad (4)$$

Lattice defects have been reported to facilitate the outward migration of excess iron ions, thereby accelerating the oxidation process of magnetite[48]. As verification, we compared the aeration oxidation kinetics of $Fe_3O_4$ NPs synthesized by two methods with different degrees of lattice defects. One was prepared by the chemical coprecipitation method as described above (Fig. 1a), which is considered to possess more lattice defects (named cc-$Fe_3O_4$ NPs). The other was prepared by the thermal decomposition method (Supplementary Fig. 12) with a relatively complete lattice structure (named TD-$Fe_3O_4$ NPs)[50]. Both $Fe_3O_4$ NPs have a similar average particle size (~10 nm) with $N(CH_3)_4{}^+$ as a surface stabilizer. Their aqueous solutions were stirred under the same aeration rate (with air) for 12 h at 120 °C. For a better comparison, the oxidation system of cc-$Fe_3O_4$ NPs (total 170 mL, 3.6 mg Fe/mL) was much larger than that of TD-$Fe_3O_4$ NPs (total 30 mL, 0.45 mg Fe/mL). This implies that individual TD-$Fe_3O_4$ could gain more oxygen than cc-$Fe_3O_4$ to keep it oxidized. From Fig. 2a, b, both $Fe_3O_4$ NPs exhibited electronic transitions in the visible and NIR region due to intervalence charge transfer between $Fe^{2+}$ and $Fe^{3+}$[51], which decreased gradually with oxidation time. At the end of aeration oxidation, little absorption beyond 700 nm was observed, indicating a phase transformation from $Fe_3O_4$ NPs to γ-$Fe_2O_3$ NPs[51]. Besides, the color of both suspensions gradually changed from dark-brown to reddish-brown. Notably, despite the less oxygen exposure for individual cc-$Fe_3O_4$ NP, its NIR absorption decreased faster than that of TD-$Fe_3O_4$ NP, especially during the initial oxidation phase (within 1 h). These results confirm that more lattice defects favor the oxidation reaction of $Fe_3O_4$ NPs due to the faster electron and ion transfer.

Analogous to aerated oxidation, the rapid electron and ion migration are also considered to facilitate the POD-like catalysis of $Fe_3O_4$ NPs, with the only difference being that the electron receptor changed from $O_2$ in aerated oxidation reaction to $H_2O_2$ in POD-like reaction. To prove this, the POD-like activity of cc-$Fe_3O_4$ NPs and TD-$Fe_3O_4$ NPs as well as their variation with aerated oxidation time were investigated. As seen in Supplementary Fig. 13, the POD-like activity of cc-$Fe_3O_4$ NPs was higher (2.8 folds) than that of TD-$Fe_3O_4$ NPs, despite TD-$Fe_3O_4$ NPs having a smaller hydrodynamic diameter and negative surface potential contributing to a strong affinity with TMB. Aeration oxidation kinetic studies show that the POD-like activity of both $Fe_3O_4$ NPs decreased with oxidation time (Fig. 2c), along with slight fluctuations in hydrodynamic size and surface potential (Supplementary Fig. 14). However, the decline rate of cc-$Fe_3O_4$ NPs was faster than TD-$Fe_3O_4$ NPs, particularly in the initial oxidation stage. This phenomenon is consistent with the changes in NIR spectra shown in Fig. 2a, b. These results further confirm that the more lattice defects of $Fe_3O_4$ NPs, the easier the migration of excess Fe ions, and thus the higher the POD-like activity. It also means that $Fe_3O_4$

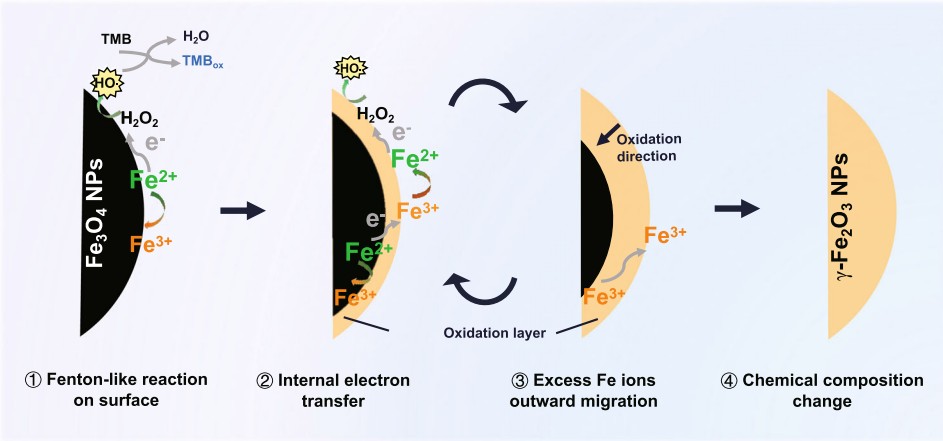

**Fig. 3 | Schematic diagram of the catalytic mechanism of the POD-like activity for Fe$_3$O$_4$ NPs.** The catalytic POD-like reaction of Fe$_3$O$_4$ NPs occurs along with internal electron transfer and excess Fe ions migration. After prolonged catalysis, Fe$_3$O$_4$ NPs suffer the phase transformation to γ-Fe$_2$O$_3$ NPs with depletable POD-like activity.

NPs with more defect sites are easier to be depleted when involved in a POD-like reaction due to their excellent catalytic capability.

## Catalytic mechanism of the POD-like for Fe$_3$O$_4$ NPs

Taken together, the catalytic mechanism of the POD-like activity for Fe$_3$O$_4$ NPs can be summarized as follows (Fig. 3): (1) fenton-like reaction on the surface. Firstly, H$_2$O$_2$ adsorbed on the surface of particles accepts electrons from the surface Fe$^{2+}$, and then dissociates into highly active ·OH to oxidize the substrates. The surface Fe$^{2+}$ is oxidized to Fe$^{3+}$. (2) Internal electrons transfer. Then, the adjacent Fe$^{2+}$ inside the surface transfers its electron to the surface Fe$^{3+}$ via the Fe$^{2+}$-O-Fe$^{3+}$ chain, retrieving the surface Fe$^{2+}$ and providing the dynamics for the sustained catalytic reaction. (3) Excess Fe ions outward migration. With the in situ oxidation of internal Fe$^{2+}$, to maintain electroneutrality, the excess Fe$^{3+}$ in the lattice has to migrate outward to the surface, leaving cation vacancies. (4) Chemical composition change. With the continuous POD-like catalytic reaction, Fe$_3$O$_4$ NPs are oxidized from the surface to the interior and finally transformed into γ-Fe$_2$O$_3$ NPs. This enzymatic-like reaction-triggered oxidation process of Fe$_3$O$_4$ NPs is thought to be analogous to the conventional low-temperature air oxidation of magnetite, in which iron ion migration is probably a rate-limiting step.

As is known, magnetite has an inverse spinel structure, with Fe$^{3+}$ occupying tetrahedral (A) sites and equal amounts of Fe$^{2+}$ and Fe$^{3+}$ occupying octahedral (B) sites, written as (Fe$^{3+}$)$_A$[Fe$^{2+}$ Fe$^{3+}$]$_B$O$_4$. The rapid electron hopping between Fe$^{2+}$ and Fe$^{3+}$ on the B-sites, creating an intermediate valence state of Fe$^{2.5+}$, contributes to the conductivity of magnetite at room temperature, exhibiting a half-metallic nature[52]. This electron-hopping process has been reported to be limited to available Fe$^{2+}$–Fe$^{3+}$ pairs and thus highly depends on the degree of non-stoichiometry of magnetite[53]. Oxidizing Fe$_3$O$_4$ to non-stoichiometry magnetite (Fe$_{3-δ}$O$_4$) or to γ-Fe$_2$O$_3$, the Fe$^{2+}$ in B-sites can be replaced by Fe$^{3+}$ and vacancies, which can be written as (Fe$^{3+}$)$_A$[Fe$_{2-6δ}$$^{2.5+}$]$_B$[Fe$_{5δ}$$^{3+}$□$_δ$]$_B$O$_4$ (□ indicates vacancy; δ indicates vacancy parameter, $0 < δ ≤ 1/3$). Thus, the number of available Fe$^{2+}$–Fe$^{3+}$ pairs decreases while isolated Fe$^{3+}$ increases. Besides, the formation of cation vacancies due to the surface migration of excess Fe$^{3+}$ can also disrupt the fast electron-hopping between Fe ions in B-sites. According to the local charge compensation model[54], each vacancy is electrically equivalent to an extra −5/2 charge at one B-site, which has to be neutralized by the excess positive charge at the adjacent B-sites. Thus, each vacancy traps 5 Fe$^{3+}$ and no longer involves in the conduction process[53]. In general, this disturbed electron-hopping process caused by the reduction of Fe$^{2+}$–Fe$^{3+}$ pairs and the formation of cation vacancies is thought to impair the

electron transfer to the surface when Fe$_3$O$_4$ nanozymes participate in the sustained POD-like reaction, leading to their depletable catalytic activity.

## LiFePO$_4$ NPs as an ideal verification model

To test the above mechanism, we found an ideal model material, LiFePO$_4$, which is commonly applied as cathode material for lithium-ion batteries (LIBs)[55–58]. LiFePO$_4$ undergoes redox reactions along with the lithium insertion/extraction during the charge-discharge process (Eqs. (5) and (6)) without changing its ordered-olivine structure (Fig. 4a)[55]. We speculate that the charging process of LiFePO$_4$ resembles the oxidation process of Fe$_3$O$_4$, both of which involve the oxidation of Fe$^{2+}$ and the migration of internal ions, which motivated us to focus on whether LiFePO$_4$ NPs also have the POD-like catalytic ability.

$$\text{LiFePO}_4 - x\,\text{Li}^+ - x\,\text{e}^- \rightarrow x\,\text{FePO}_4 + (1-x)\text{LiFePO}_4 \qquad (5)$$

$$\text{FePO}_4 + x\,\text{Li}^+ + x\,\text{e}^- \rightarrow x\,\text{LiFePO}_4 + (1-x)\text{FePO}_4 \qquad (6)$$

Rod-like LiFePO$_4$ NPs with an average length of 321.9 nm and width of 172.2 nm (Fig. 4b) were successfully synthesized using the solvothermal method[59] and characterized via various methodologies (Supplementary Fig. 15 and Supplementary Tables 2 and 3). As expected, the POD-like activity of LiFePO$_4$ NPs was demonstrated with different chromogenic substrates including TMB, ABTS, and OPD (Fig. 4c and Supplementary Fig. 16). Also, they follow pH, temperature as well as NPs concentration dependence, and the Michaelis−Menten kinetics (Supplementary Figs. 17 and 18). The optimal pH is about 4.0. The ·OH was shown to be generated in a time- and concentration-dependent manner (Fig. 4d and Supplementary Fig. 19), which is similar to Fe$_3$O$_4$ NPs. We then compared the POD-like activity of LiFePO$_4$ NPs and cc-Fe$_3$O$_4$ NPs using two oppositely charged substrates (TMB and ABTS) at pH 3.6. The results consistently show that LiFePO$_4$ NPs had higher catalytic ability than cc-Fe$_3$O$_4$ NPs (Supplementary Fig. 20), and the $a_{nano}$ of LiFePO$_4$ NPs was approximately four times than that of cc-Fe$_3$O$_4$ NPs, despite their larger particle size (Fig. 4e). These results imply that LiFePO$_4$ NPs may share a similar POD-like catalytic mechanism with Fe$_3$O$_4$ NPs, differing in that the rapid Li$^+$ migration in the lattice of LiFePO$_4$ NPs confers them a superior POD-like catalytic activity (Fig. 4f).

## Phase transformation of LiFePO$_4$ NPs

The recycled LiFePO$_4$ NPs from three cycles of POD-like catalysis were proven to suffer a phase transformation to FePO$_4$ via multiple characterization techniques. Specifically, the XPS Fe2p peaks of the

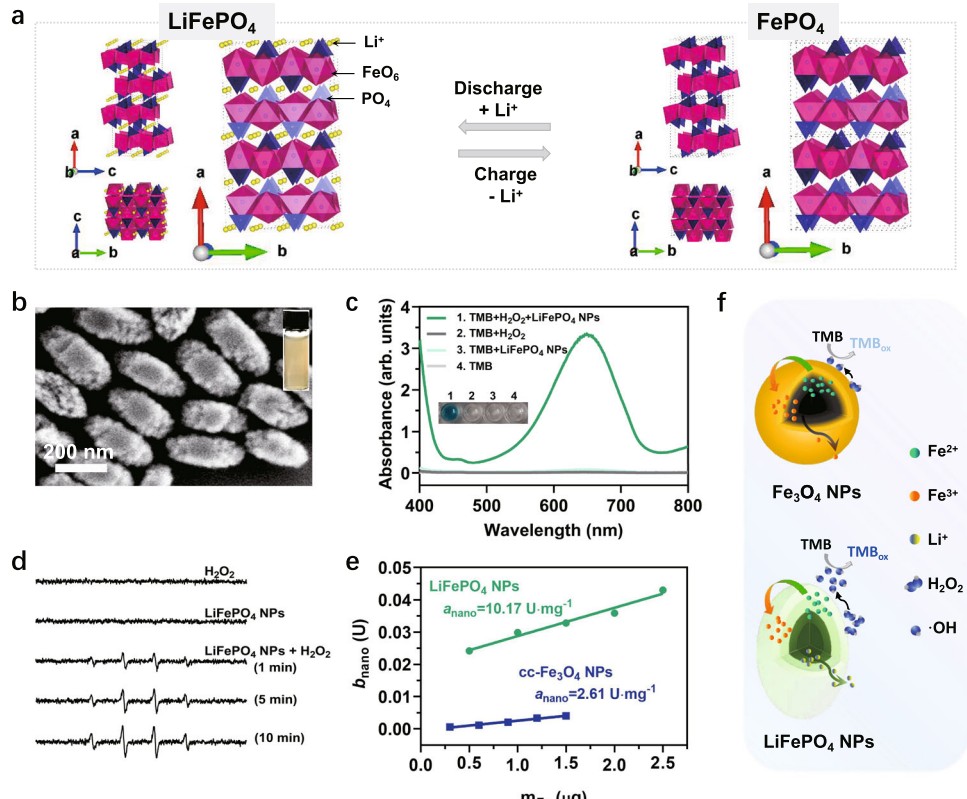

**Fig. 4 | LiFePO₄ NPs as verification materials and their POD-like activity. a** The crystal structure of LiFePO₄ and FePO₄ viewed along the *a*, *b*, *c*-axis. The olivine structure is maintained during Li-ions insertion and extraction. **b** The SEM image of as-synthesized LiFePO₄ NPs. The image was collected at least three times. Inset is a photo of LiFePO₄ NPs aqueous solution. **c** The POD-like activity of LiFePO₄ NPs (6.25 µg Fe/mL) with TMB (1.7 mM) as colorimetric substrates under the presence of $H_2O_2$ (0.8 M) in 0.2 M acetate buffer (pH = 3.6). **d** ESR spectra of spin adducts DMPO/·OH produced by LiFePO₄ NPs (10 µg/mL) in the presence or absence of $H_2O_2$ (0.165 M) in 0.2 M acetate buffer (pH = 3.6). **e** Comparison of the $a_{nano}$ of as-synthesized LiFePO₄ NPs and cc-Fe₃O₄ NPs. Error bars represent standard deviation from three independent measurements. **f** Diagram of the POD-like catalytic reaction process of LiFePO₄ NPs and Fe₃O₄ NPs.

recycled LiFePO₄ NPs were shifted toward the higher binding energy (Fig. 5a), indicating the oxidation of $Fe^{2+}$ within the NPs. In the XRD pattern (Fig. 5b), the residual LiFePO₄ phase (marked with o in the yellow pattern) in the recycled NPs was negligible, proving that almost all LiFePO₄ were delithiated and oxidized into FePO₄ (marked with +) after cyclic POD-like catalysis. This result was further confirmed by ICP analysis that the Li element content in recycled NPs was almost 0 (Table 1). This phase transformation, as expected, severely impaired the POD-like activity of the recycled LiFePO₄ NPs (Fig. 5c), in agreement with the self-depleting characteristic of the Fe₃O₄ NPs described above.

As a LIBs cathode material, the reversible lithiation and delithiation characteristic of LiFePO₄ contribute to its superior electrochemical performance. We measured the cyclic voltammograms (CV) of the as-prepared LiFePO₄ NPs before and after participating in the cyclic POD-like reactions under different scan rates in the electrolyte containing $Li^+$ or $Na^+$. As shown in Supplementary Fig. 21, the increase in redox peak currents ($I_p$) was proportional to the square root of scan rate ($v^{1/2}$), implying a diffusion-controlled process of $Li^+$ or $Na^+$ extraction and insertion[60]. Noticeably, the $I_p$ of the recycled LiFePO₄ NPs (i.e., FePO₄) was obviously reduced compared to LiFePO₄ NPs (Fig. 5d), especially in the electrolyte containing $Na^+$ (Supplementary Fig. 21e–h), indicating that the presence of mobile $Li^+$ contributes significantly to the redox peak current. As a comparison with LiFePO₄ NPs, the CV curves of Fe₃O₄ NPs were also measured under the same conditions. As shown in Supplementary Fig. 22, the $I_p$ also exhibits a linear relation with the $v^{1/2}$. However, unlike LiFePO₄ NPs, the $I_p$ of the recycled Fe₃O₄ NPs (i.e., γ-Fe₂O₃) only showed a marginal decrease compared to their counterparts before participating in the catalytic reaction, both of

which were found to be similar to the $I_p$ of recycled LiFePO₄ NPs under the same scanning rate (Supplementary Fig. 22 and Fig. 5d). This is probably explained by the lack of freely diffusing ions in the lattice of iron oxide and FePO₄, which weakens the electron transfer process in their redox reactions. By contrast, LiFePO₄ NPs exhibited the highest $I_p$, ascribed to the availability of Li ions in their crystals.

## Mobile Li-ions as the limiting factor

In the field of sodium (Na)-ion batteries, the charge transfer resistances and lattice volume change upon $Na^+$ migration are larger for NaFePO₄ electrodes, compared with their Li equivalents due to the larger ionic radius of Na (1.02 Å) than Li (0.76 Å)[61]. Inspired by this, We partially replaced Li with Na in the lattice of LiFePO₄ NPs to explore the potential effect of Na-doping on their POD-like activity. Concretely, three NaLiFePO₄ NPs with similar physicochemical properties but different Na-doping amounts were successfully synthesized (Supplementary Fig. 23 and Supplementary Table 4). We then compared their POD-like activities under the same reaction conditions and found that the more Na doping, the lower the POD-like activity (Fig. 5e), indicating that the large $Na^+$ radius hinders the free migration of $Na^+$ and $Li^+$ in the crystal, thereby impairing the electron transfer rate. We attempted to use K-doped LiFePO₄ NPs as further proof, however, the large ionic radius of K (1.38 Å) makes it difficult to embed into the electrode materials (Supplementary Fig. 24 and Supplementary Table 4), which is a common issue in K-ion batteries[62].

To further prove the decisive role of mobile $Li^+$, we measured the POD-like activity of commercially available LiFePO₄, Fe₃(PO₄)₂, and FePO₄ materials with similar hydrodynamic dimensions and surface negative potentials (Supplementary Fig. 25). The results show that

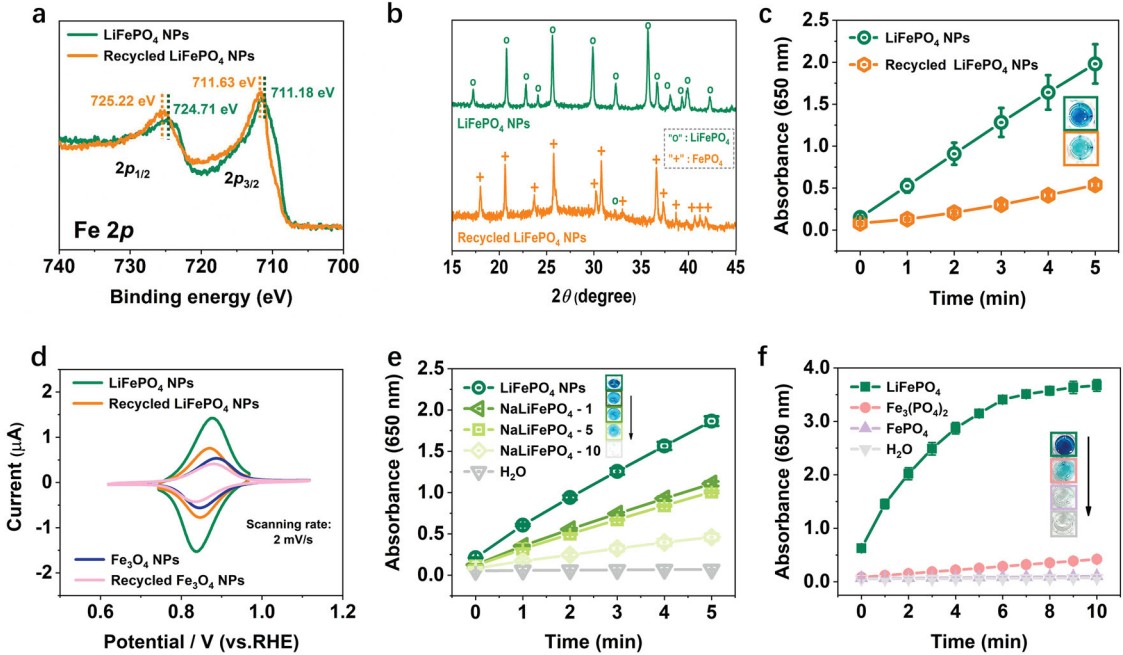

**Fig. 5 | Catalytic mechanism of the POD-like activity for LiFePO₄ NPs. a** The Fe2*p* XPS spectra, **b** XRD patterns, **c** POD-like activities of the control LiFePO₄ NPs and the recycled LiFePO₄ NPs from three cycles of POD-like reaction. **d** The CV curves of LiFePO₄ NPs and recycled LiFePO₄ NPs in lithium acetate buffer solution, compared with that of the Fe₃O₄ NPs and recycled Fe₃O₄ NPs. Scanning rate is 2 mV/s. **e** Comparison the POD-like activity of NaLiFePO₄ NPs (0.4 μg Fe/mL) with different Na-doping levels. **f** Comparison the POD-like activity of commercial LiFePO₄, Fe₃(PO₄)₂, and FePO₄ (1.25 μg Fe/mL) using TMB (1.7 mM) as substrates in the presence of H₂O₂ (0.8 M) at pH 3.6. Error bars represent standard deviation from three independent measurements.

their POD-like activity follows LiFePO₄ >> Fe₃(PO₄)₂ > FePO₄ (Fig. 5f), directly confirming that the presence of Fe²⁺ alone in Fe₃(PO₄)₂ cannot ensure the superior catalytic performance, but the transportable Li⁺ contributes to the outstanding POD-like activity of LiFePO₄.

In summary, the catalytic mechanism of the POD-like activity of Fe₃O₄ nanozymes is elucidated by characterizing the chemical composition and catalytic activity of the Fe₃O₄ NPs recycled from the long-term POD-like catalysis. These studies indicate that not only the surface Fe²⁺, but also the internal Fe²⁺ contribute to the POD-like activity of Fe₃O₄ nanozymes. The Fe²⁺ inside the particle can transfer electrons to the surface, regenerating the surface Fe²⁺ that is constantly involved in the sustained catalytic reaction. This process is usually accompanied by the outward migration of excess oxidized Fe³⁺ from the interior of the crystal, which is considered as a rate-limiting step. Analogous to the low-temperature oxidation of magnetite, Fe₃O₄ NPs participated in the POD-like reaction are eventually oxidized to γ-Fe₂O₃ NPs with a reduced POD-like capacity. Furthermore, this mechanism is well-validated on LiFePO₄ NPs. This work reveals the depletable property of Fe₃O₄ nanozymes differing from natural enzymes and highlights the potential contribution of internal metal atoms in nanozymes-catalyzed reactions. Meanwhile, these findings provide a theoretical basis for the mechanistic study and rational design of other related nanozymes.

**Table 1 | ICP analysis of the control and recycled LiFePO₄ NPs**

| Element | Concentration (μg/mL) | |
|---|---|---|
| | Control LiFePO₄ NPs | Recycled LiFePO₄ NPs |
| Li | 15.38 | 0.66 |
| Fe | 120.50 | 126.54 |
| P | 71.02 | 63.00 |
| Molar ratio of Li:Fe:P | 0.97:0.94:1 | 0.05:1.11:1 |

## Methods

### Chemicals

Ferrous sulfate heptahydrate (FeSO₄·7H₂O), iron (III) chloride hexahydrate (FeCl₃·6H₂O), dimethyl sulfoxide, TMB (≥99%), ABTS (98%), OPD (98%), ferric oxide (Fe₂O₃), iron acetylacetonate [Fe(acac)₃], oleic acid (OA), dibenzyl ether, lithium hydroxide (LiOH·H₂O), ferric phosphate (FePO₄), rhodamine B, terephthalic acid (TA), sodium carbonate (Na₂CO₃), and sodium acetate (CH₃COONa) were purchased from Aladdin (Shanghai, China). Tetramethylammonium hydroxide (TMAOH) and 5,5-dimethyl-1-pyrroline N-oxide (DMPO) were purchased from Sigma Aldrich. Hydrochloric acid (HCl, 36.0–38.0%), nitric acid (HNO₃), phosphoric acid (H₃PO₄, 85% wt), acetic acid (CH₃COOH), hydrogen peroxide (H₂O₂, 30%), ethyl alcohol, ethylene glycol (EG), and potassium hydroxide (KOH) were purchased from Sinopharm Chemical Reagent Co., Ltd. Ferrous phosphate [Fe₃(PO₄)₂] was purchased from Shanghai Maclin Biochemical Technology Co., Ltd. Lithium iron phosphate (LiFePO₄) was purchased from Shanghai Xushuo Biological Technology Co., Ltd. All chemicals were used as received without further purification. Deionized water was used throughout the experiments.

### Characterization

The particle size and morphology were observed by scanning electron microscopy (SEM, Ultra Plus, Carl Zeiss, Germany). TEM, selective area electron diffraction (SAED), and high-resolution TEM (HRTEM) were taken using transmission electron microscopy (TEM, JEOL JEM-2100F, Japan). The phase and crystalline structure were explored by XRD (Brucker D8 Advance, Germany) and the DIFFRAC plus XRD Commander software (v2.6.1), using Cu Kα radiation. The element composition, chemical structures, and properties of NPs were analyzed by SEM energy dispersive spectroscopy (EDS), ICP-OES coupled with ICP Expert II software (Agilent ICPOES730), Fourier transform infrared spectroscopy coupled with OMNIC software v9.6 (FTIR, Nicolet IS 10, Thermo, USA), Raman spectrometer coupled with WiRE™ software v2.0

(Renishaw Invia, UK) and X-ray photoelectron spectroscopy coupled with Avantage software v5.967 (XPS, Thermo ESCALAB 250Xi, USA). The chemical speciation of Fe in IONPs was determined by NEXAFS at the beamline of 08U1A in the Shanghai Synchrotron Radiation Facility, Shanghai, China. The NEXAFS spectra of the references including $FeSO_4$ and $Fe_2O_3$ were also measured. Hydrodynamic diameters and Zeta potential of particles were measured by the dynamic light scattering (DLS, Nano-ZS90, Malvern, England) and Malvern Zetasizer software (v7.12). The UV-Vis-NIR absorption spectra were measured using UV3600 (Shimadzu, Japan) or Microplate Reader (Infinite M200, Tecan, Switzerland). CV was measured using Electrochemical Workstation (CHI760e) and CHI-760E electrochemical software (v20.04). The production of ·OH was detected by the electron spin resonance spectrometer (ESR, Bruker EMXplus, Germany) and Bruker Xenon software (v1.2) at ambient temperature.

### Synthesis of IONPs by chemical coprecipitation method

The naked magnetite NPs were synthesized by a modified chemical coprecipitation method, namely, cc-$Fe_3O_4$ NPs[24]. In brief, 2.7025 g of $FeCl_3$·$6H_2O$ and 1.39 g $FeSO_4$·$7H_2O$ were added to a 100 mL three-necked flask with 10 mL of 2 M HCl reaction solvent. The mixture was stirred at 400 rpm for 10 min under a nitrogen gas ($N_2$) atmosphere. Then, 62.5 mL of 12.5% TMAOH was quickly poured into the above flask and stirred at 700 rpm for 1 h at room temperature. $N_2$ was introduced throughout the experiment to remove $O_2$ from the reaction system. The reaction solution was transferred to a beaker and washed with deoxygenated water three times by magnetic decantation to remove the residual reagents. Part of obtained precipitate was dispersed into deionized water with a pH of 3 to obtain the cc-$Fe_3O_4$ NPs suspensions, which were stored at 4 °C for subsequent experiments.

The remaining cc-$Fe_3O_4$ NPs precipitates were dried into black $Fe_3O_4$ NPs powder under vacuum, and then calcined at 200 and 650 °C for 2 h to obtain γ-$Fe_2O_3$ NPs and α-$Fe_2O_3$ NPs powder, respectively[34]. Finally, these three powders were dissolved in a pH of 3 aqueous solutions by ultrasound and stored at 4 °C for subsequent experiments.

### Synthesis of $Fe_3O_4$ NPs by thermal decomposition method

Firstly, the OA-coated $Fe_3O_4$ NPs were synthesized according to the thermal decomposition method (denoted as TD-$Fe_3O_4$@OA NPs)[50]. The chloroform solvent was removed from 4 mL of TD-$Fe_3O_4$@OA NPs (1.627 mg Fe/mL) by rotary evaporation. Then, the OA layer on the particle surface was replaced with 15 mL of 0.5% TMAOH under ultrasonic stirring[63]. TMAOH is a phase transfer small molecule agent which can form an electrostatic double layer on the particle surface and stabilize the particles in the aqueous phase. Next, the reaction system was transferred to 60 mL of the separatory funnel, and 45 mL of chloroform was added. The mixture was thoroughly mixed. After standing for 15 min, the underlying oil phase liquid was discarded. This extraction procedure was repeated three times to fully remove the free OA that was replaced from the particle surface by TMAOH. Finally, the obtained $Fe_3O_4$ NPs aqueous solution (denoted as TD-$Fe_3O_4$ NPs) was filtered through a 0.22-μm filter and stored at 4 °C for subsequent experiments.

### Synthesis of $LiFePO_4$ NPs and Na- or K-doped $LiFePO_4$ NPs

$LiFePO_4$ NPs were successfully synthesized by glycol-based solvothermal method[59]. In a typical route, 0.27 mL $H_3PO_4$ (85% wt) was dropped into 9 mL of EG containing 0.365 g of LiOH·$H_2O$. The mixture was stirred thoroughly until the neutralization reaction was complete, showing a milk-white suspension. Then 9 mL of EG dissolved 0.890 g $FeSO_4$·$7H_2O$ was added into the suspension under vigorous stirring. The mixtures slowly turned grayish-green. After stirring for 30 min, the

reaction solution was transferred into a sealed hydrothermal reactor, and heated at 180 °C for 10 h. At the end of the reaction, the final products were cooled down to room temperature and washed with deionized water three times.

For Na or K-doped $LiFePO_4$ NPs, the Li element content (8.69 mmol) of 1%, 5%, and 10% in the above method was replaced with Na element in $Na_2CO_3$ or K element in KOH. The rest of the synthesis steps remained unchanged. Briefly, for Na-doped $LiFePO_4$ NPs, 0.365 g of LiOH·$H_2O$ was replaced with 4.6 mg of $Na_2CO_3$ and 0.361 g of LiOH·$H_2O$ (denoted as NaLiFePO$_4$–1), 23.0 mg of $Na_2CO_3$ and 0.346 g of LiOH·$H_2O$ (denoted as NaLiFePO$_4$–5), and 46.1 mg of $Na_2CO_3$ and 0.328 g of LiOH·$H_2O$ (denoted as NaLiFePO$_4$–10), respectively. For K-doped $LiFePO_4$ NPs, 0.365 g of LiOH·$H_2O$ was replaced with 4.9 mg of KOH and 0.361 g of LiOH·$H_2O$ (denoted as KLiFePO$_4$–1), 24.3 mg of KOH and 0.346 g of LiOH·$H_2O$ (denoted as KLiFePO$_4$–5), and 48.7 mg of KOH and 0.328 g of LiOH·$H_2O$ (denoted as KLiFePO$_4$–10), respectively.

### The POD-like activity of NPs

The POD-like activity of NPs was measured by different colorimetric substrates, including TMB, ABTS, and OPD, in the presence of $H_2O_2$ under different reaction conditions. The absorbance of the colored oxidation products at the corresponding wavelength (TMB$_{ox}$: 650 nm; ABTS$_{ox}$: 415 nm; OPD$_{ox}$: 492 nm) was monitored by a microplate reader and Tecan i-control software (v1.6.19.2). The effects of pH (2.5–11), temperature (10–65 °C), and $LiFePO_4$ NPs concentration (0–6.25 μg Fe/mL) on POD-like activity were also studied.

### Measurement of the specific activity ($a_{nano}$) of NPs

The POD-like specific activity of NPs synthesized in this work was determined according to the modified method specified in the national standard of China (GB/T 37966-2019)[35]. The following provided the general procedures: (a) monitor the temperature inside the quartz cuvette and incubate all reagents and samples to 25 °C; (b) add 2.000 mL of 0.2 M acetate buffer solution (pH = 3.6) to a reaction container; (c) add 0.100 mL of different concentrations of NPs and 0.100 mL of 10 mg/mL of TMB in order, mix completely and incubate for 60 s at 25 °C; (d) add 0.200 mL of 30% $H_2O_2$ and mix completely; (e) immediately transfer the appropriate amount of reaction solution to the cuvette and monitor the changes of absorbance at 650 nm within the specified times using UV-Vis spectroscopy and UV Probe software (v2.42). The initial change rate of absorbance (min$^{-1}$) is obtained from the slope of the early phase of time course; (f) measurement of reagent blank rate 1 and reagent blank rate 2: 30% $H_2O_2$ and NPs are respectively replaced by deionized water. The measurement procedure is as described in Step a–e above; (g) after deducting the reagent blank rate 1 and 2, the POD-like catalytic activity units ($b_{nano}$) of NPs can be calculated according to Eq. (7):

$$b_{nano} = \frac{V \times (\triangle A / \triangle t)}{\varepsilon \times l} \qquad (7)$$

where $b_{nano}$ is the POD-like catalytic activity units of NPs (U); V is the total volume of reaction solution (μL); ΔA/Δt is the slope of the initial liner portion of absorbance changing over time after correcting with reagent blank rate 1 and 2 (min$^{-1}$); $\varepsilon$ is the molar absorption coefficient of TMB derivative (39,000 mol$^{-1}$·L·cm$^{-1}$); l is the optical path of the cuvette (cm).

The specific activity of NPs ($a_{nano}$) can be calculated according to Eq. (8):

$$a_{nano} = \frac{\triangle b_{nano}}{\triangle m_{Fe}} \qquad (8)$$

where $a_{nano}$ is the specific activity of NPs ($\text{U·mg}^{-1}$); $m_{Fe}$ is the total Fe element mass contained in added NPs (mg); $\frac{\triangle b_{nano}}{\triangle m_{Fe}}$ is the slope of the curve plotting the $b_{nano}$ against different masses of NPs ($m_{Fe}$).

### ESR measurement

The production of ·OH was detected by adding 10 µg Fe/mL of cc-$Fe_3O_4$ NPs or $LiFePO_4$ NPs and 30 mM of DMPO into 300 µL of acetate buffer (pH = 3.6) with or without 0.5% $H_2O_2$. The signal of the spin adduct (DMPO/·OH) was recorded at ambient temperature at the 1st, 5th, and 10th min after all reagents were mixed. The experimental parameters were as follows: 1 G modulation amplitude, 100 kHz modulation frequency, 6.325 mW microwave power, 9.829 GHz resonance frequency.

### ·OH detection using terephthalic acid as a fluorescent probe

TA is highly selective and can capture ·OH generated in situ, generating 2-hydroxyterephthalic acid (TAOH) with unique fluorescence around 534 nm, which can be detected by fluorescence spectrophotometer and FluorEssence software (v3.8). In this work, 0.2 M $H_2O_2$ was added to 0.2 M acetate buffer (pH = 3.6) containing nanozymes (different materials and concentrations) and 0.5 mM TA. After mixing, the fluorescence signal was detected on a fluorescence spectrophotometer at specific times in the range of 350–600 nm with an excitation wavelength of 315 nm.

### Cyclic POD-like catalysis of NPs

For IONPs, 0.1 g of $Fe_3O_4$ NPs or γ-$Fe_2O_3$ NPs powder was added to a 400 mL of acetate buffer (pH 3.6) containing 20 mL TMB (10 mg/mL) and 40 mL of 30% $H_2O_2$. This catalytic reaction system lasted for one day (20 h) at 25 °C. After catalysis, $Fe_3O_4$ NPs or γ-$Fe_2O_3$ NPs in the reaction solution were recycled by magnetic separation, and then ultrasonic washed with deionized water several times. The recycled NPs were re-added into a new reaction system mentioned above to catalyze a new round of POD-like reactions. Totally, 5 rounds (or days) of catalysis were carried out, and each round lasted for 20 h at 25 °C. The $Fe_3O_4$ NPs recycled on days 0, 1, 3, and 5 were taken for further characterization.

For $LiFePO_4$ NPs, 1 mL of $LiFePO_4$ NPs (8.4 mg Fe/mL) was added to a 20 mL of acetate buffer (pH 3.6) containing 1 mL TMB (10 mg/mL) and 2 mL of 30% $H_2O_2$. This catalytic reaction system lasted for 1 h at 25 °C. After catalysis, $LiFePO_4$ NPs in the reaction solution were recycled by centrifugation, and then ultrasonic washed with deionized water several times. The recycled $LiFePO_4$ NPs were re-added into a new reaction system mentioned above to catalyze a new round of POD-like reactions. Totally, 3 rounds of catalysis were carried out, and each round lasted for 1 h at 25 °C. The $LiFePO_4$ NPs recycled from the 0 and 3 rounds were taken for further characterization.

### Aerated oxidation of $Fe_3O_4$ NPs

A total of 170 mL of the as-synthesized cc-$Fe_3O_4$ NPs suspensions (3.57 mg Fe/mL) was transferred into a 250 mL three-necked flask and heated from room temperature to 120 °C. Air was pumped into the suspension at a constant rate (9 L/min) for 12 h under stirring (350 rpm). Condensation reflux was kept throughout the aeration oxidation. Samples oxidized at 0, 0.5, 1, 3, 5, 8 10, and 12 h were used for subsequent characterization.

As for TD-$Fe_3O_4$ NPs, the only difference was that 30 mL of TD-$Fe_3O_4$ NPs (0.45 mg Fe/mL) were transferred into a 100 mL three-necked flask. Other steps were consistent with the aeration oxidation of cc-$Fe_3O_4$ NPs.

### Electrochemical performance of NPs

The CV was used to evaluate the electrochemical performance of NPs in a three-electrode cell[35]. A total of 10 µL of NPs (100 µg Fe/mL) was mixed with 90 µL of 0.5% Nafion solution, and then dropped 30 µL of the above mixture into the polished glassy carbon electrode, which was dried and used as the working electrode. Mercury/mercurous sulfate (Hg/$Hg_2SO_4$) was used as the reference electrode and platinum wire as the counter electrode. The electrolyte solution was 0.2 M lithium acetate or sodium acetate buffer (pH = 3.6). The CV was recorded at different voltage scan rates (v).

### Statistical and reproducibility

All experiments were repeated three times independently, and the data were presented as the mean values ± standard deviations (SD). Error bars shown in this paper represent the SD derived from three repeated measurements. UV probe Software v2.42, Tecan i-control Software v1.6.19.2, Malvern Zetasizer Software v7.12, FluorEssence Software v3.8, Bruker Xenon Software v1.2, CHI-760E electrochemical Software v20.04, WiRE™ 2.0 Software, Thermo Avantage Software v5.967, ICP Expert II Software, OMNIC Software v9.6, and DIFFRAC plus XRD Commander Software v2.6.1 were used for data collection. GraphPad Prism 8 and Origin 2018 were used for data processing and statistical analysis.

### Reporting summary

Further information on research design is available in the Nature Research Reporting Summary linked to this article.

## Data availability

All data supporting the findings of this study are available within the article and its Supplementary Information files. Data are available from the corresponding author upon request. Source Data are provided with this paper.

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

## Acknowledgements

This work was supported by the National Key Research and Development Program of China (2017YFA0205502), National Natural Science Foundation of China (82072067 and 61821002), and the Fundamental Research Funds for the Central Universities.

## Author contributions

Y.Z. conceived and supervised the project; H.D. and Y.Z. designed the studies; H.D. performed the main experiments and wrote the manu-script; W.D. helped with the data analysis. J.D. performed the Raman experiments; R.C. performed the electron microscopy experiments; F.K. provided the TD-Fe$_3$O$_4$ nanoparticles; W.C., M.M., and N.G. helped with article revisions. All the authors discussed the results and commented on the manuscript.

## Competing interests

The authors declare no competing interests.
