## [Peer Review File · Nature Communications]

Depletable Peroxidase-like Activity of Fe₃O₄ Nanozymes Accompanied with Separate Migration of Electrons and Iron IonsREVIEWER COMMENTS

Reviewer #1 (Remarks to the Author):

As main results, the manuscript gives the oxidation process of the Fe₃O₄ during the POD-like activity in a time range larger than the expected to maghemite follow by XPS and Raman is an interesting approach to study the problem, presenting interesting results such as the similitudes to the temperature-induced oxidation process and the Fe³⁺ diffusion involved.

Considering the impact of the Journal aimed by the authors, a key point is the relevance of the subject and the originality of the conclusions are also very important, together with DATA and Procedure strongness. In a general way, it is a very interesting article with interesting results and an important question to be analyzed: the mechanism of peroxidase-like activity of Fe₃O₄ nanoparticles in an "atomic level". The data presented is robust, at least apparently, and the methodology used in the nanoparticle synthesis is usual, as well the procedures and techniques used characterize the Nanoparticle's characterization. The main objectives planted in the manuscript, dealing with "...the internal atomic changes and their contribution to the catalytic reaction..." of Fe₃O₄ nanoparticles is of great interest for a broad audience. In fact, the mimetic enzymatic nature of the catalytic activity addressed to ferrite nanoparticle stills itself in discussion. Consequently, studies on the mechanism at atomic level acting in this catalytic activity are of great interest, with really fell works dealing with it. The statements "...a detailed mechanism of the POD-like activity of Fe₃O₄ nanozymes is elucidated...", "...demonstrate that all Fe²⁺ in Fe₃O₄ nanozymes contribute to their POD-like activity." and "The Fe²⁺ inside the particle transfers electrons to the surface, regenerating the surface Fe²⁺ that is directly involved in the sustained catalytic reaction." are really strongly, requesting strong evidences.

The article is comprehensive and clearly presented.

However, in the introduction, the authors present the peroxidase-like activity of the Fe₃O₄ nanoparticles as unquestionable. However, there are discrepancies, with interesting alternating analyses. For clarity, these alternative points should be presented in the introduction with corresponding references (see: Michel Nguyen, Anne Robert, Bernard Meunier. "Nanozymes" without catalytic activity should be renamed. (2021) hal-03290323) .

The methodology presented to addresses the main question of the work is interesting. Firstly, the use of the physicochemical analysis of the recycled of the nanoparticles after "POD"-activity is apparently solid.

In a general way, the article is very interesting and has a strong potential to be published. However, in my opinion, several points should be clarified previously.

- Firstly, the "POD"-activity was performed in a really acid pH (3.6). A question rapidly arises: Is a Fe leaching process taking place at this pH? It may be a critical issue due to the estimation and the mechanism acting in the oxidation reactions observed. I suggest to the author using magnetic separation (both more interesting ferrites present strong magnetic response) in the solution of acid buffer to analyze the presence of Fe ions.

- In addition, the Fe³⁺ migration to the internal zones of the nanoparticles and the electron transference to the surface are concluded by comparing the Fe₃O₄ oxidation results and the comparison with the "witness" LiFePO₄ results. However, in my opinion, more direct evidence should be presented. XPS and RAMAN give a global view of the particle oxidation. I suggest to use an EELS mapping along the nanoparticles exposed to POD-activity after different times in a TEM, taking advantage from the spectral differences of Fe²⁺/Fe³⁺.

- In my opinion, a more detailed analysis on the "electron transference" should be presented. Is the electron hope in B site, or the intervalence, acting here? Is the half-metallic nature of magnetite arising from it a key point here? If it is correct, how the migration of Fe³⁺ and the gradual oxidation to maghemite, an isolator, affect this mechanism, since the intervalence is an exclusive feature of magnetite within the iron-oxides family, being depleted with the oxidation.

- Finally, why was the electrochemical characterization performed only in the LiFePO₄ system and not in the Fe₃O₄? Li also presents a complex chemistry, forming oxidizing radicals with oxygen atoms, which is use for isotopic separation, for example. In order to be use as a "witness" system, I expect a comparative study of the electrochemical response of the Fe₃O₄ nanoparticles.

In addition, small observations arise from the methodology. The nanoparticles prepared by thermal decomposition are presented as “naked-ones”, however, a phase transfer is performed (OA => TMAOH). Are the particles “naked”? More details about this should be given in the Materials and Methods. Another small point concerns the EPR measurements at very low reaction times. Firstly, when do the authors consider the reaction start point? Second, how did the authors to measure the ESR after 1 min, and even 5 min., taking into account that is necessary to center the sample in the cavity, to measure in the critical coupled condition with the cavity (or matched) and to perform the field scanning (was used only one scan?).

In my opinion, the article has potential to be published, and the subject and the approach are very interesting. However, I think that the points listed above should be addressed, especially the first two.

Reviewer #2 (Remarks to the Author):

The author detailed the mechanism of the POD-like activity of Fe₃O₄ nanozymes and demonstrate that all Fe²⁺ in Fe₃O₄ nanozymes contribute to their POD-like activity. The Fe²⁺ inside the particle transfers electrons to the surface, regenerating the surface Fe²⁺ that is directly involved in the sustained catalytic reaction. An interesting work, however still few question remain to establish this mechanism.

- (1) Though authors showed the transformation of Fe²⁺ to Fe³⁺ with different time duration and XPS characterization, however still the mechanism for contributing inside Fe²⁺ in their POD like activity.
- (2) Leaching effect still exists, in that case, the nanozyme can't be re-use in real-life application. This thing should be considered.
- (3) Phase dependent activity like why α and γ iron oxide proving different POD-like activity is also missing.
- (4) Few recently published articles (e.g. <https://doi.org/10.1002/adma.202107088>; <https://doi.org/10.1002/ange.202112453>; <https://doi.org/10.1039/C9TB00989B>) should be discussed and cited.

Reviewer #3 (Remarks to the Author):

The manuscript "The Depletable Peroxidase-like Activity of Fe₃O₄ Nanozymes Accompanied with Phase Transformation Triggered by Separate Migration of Electrons and Iron Ions" by Deng et al. reports on an aspect of the catalytic activity of Fe₃O₄ nanozymes, namely the electron transfer via a Fe(2+)-O-Fe(3+) chain leading to a reduction of iron charge at the NP surface. This highlights the importance of bulk-surface atom interaction in the catalytic reaction.

Overall, the findings are interesting and definitely relevant for the research community focusing on catalytic materials. However, the focus of the work is quite narrowly set on the specific aspect of surface Fe(2+) activation and the impact of surface passivation on catalytic activity. The authors suggest a possible mechanism that leads to passivation of the nanozyme without providing any alternative for how this could be remediated. This focus makes the work very limited with respect to applicability and possible pathways of improving the performance of catalysts. Although the electron transfer mechanism is intriguing in itself and the comparison with LiFePO₄ makes a strong point, I see a lack of generality that would make this work worthy of being published in Nature Communications.

Therefore, I suggest considering this work for a more specialised journal.

Minor comments:

Figs. 1(e) and (h) are too small. Details a very hard to identify.

Label of y-axis in Fig. 1(b) is "bnano" and should be "anano"

Responds to the reviewer's comments:

Reviewer #1 (Remarks to the Author):

As main results, the manuscript gives the oxidation process of the Fe₃O₄ during the POD-like activity in a time range larger than the expected to maghemite follow by XPS and Raman is an interesting approach to study the problem, presenting interesting results such as the similitudes to the temperature-induced oxidation process and the Fe³⁺ diffusion involved. Considering the impact of the Journal aimed by the authors, a key point is the relevance of the subject and the originality of the conclusions are also very important, together with DATA and Procedure strongness. In a general way, it is a very interesting article with interesting results and an important question to be analyzed: the mechanism of peroxidase-like activity of Fe₃O₄ nanoparticles in an "atomic level". The data presented is robust, at least apparently, and the methodology used in the nanoparticle synthesis is usual, as well the procedures and techniques used characterize the Nanoparticle's characterization. The main objectives planted in the manuscript, dealing with "...the internal atomic changes and their contribution to the catalytic reaction..." of Fe₃O₄ nanoparticles is of great interest for a broad audience. In fact, the mimetic enzymatic nature of the catalytic activity addressed to ferrite nanoparticle stills itself in discussion. Consequently, studies on the mechanism at atomic level acting in this catalytic activity are of great interest, with really fell works dealing with it.

Response. Thank you very much for your positive comments.

The statements "...a detailed mechanism of the POD-like activity of Fe₃O₄ nanozymes is elucidated...", "...demonstrate that all Fe²⁺ in Fe₃O₄ nanozymes contribute to their POD-like activity." and "The Fe²⁺ inside the particle transfers electrons to the surface, regenerating the surface Fe²⁺ that is directly involved in the sustained catalytic reaction." are really strongly, requesting strong evidences. The article is comprehensive and clearly presented.

Response. We are extremely grateful to reviewer for pointing put this problem. We have modified the statement of the concluding sentences in the manuscript to make them slightly hedging. These revised words/sentences were listed below and were highlighted in red in the revised manuscript.

Abstract, Page 2, Line 5-8: "Here we report that Fe²⁺ within Fe₃O₄ can transfer electrons to the surface via the Fe²⁺-O-Fe³⁺ chain, regenerating the surface Fe²⁺ and enabling a sustained POD-like catalytic reaction. This process usually occurs with the outward migration of excess oxidized Fe³⁺ from the lattice, which is a rate-limiting step."

Page 5, Line 5-8: "Generally, Fe²⁺ inside the particle could transfers its electron to the surface layer, regenerating the surface Fe²⁺ and sustaining the catalytic reaction. This process is usually coupled with the outward migration of excess oxidized Fe³⁺, which is probably a rate-limiting step."

Page 12, Line 12-13: “Analogous to aerated oxidation, the rapid electron and ion migration is also considered to facilitate the POD-like catalysis of Fe₃O₄ NPs, with the only difference that the electron receptor changed from O₂ in aerated oxidation reaction to H₂O₂ in POD-like reaction.”

Page 14, Line 1-2: “the excess Fe³⁺ in the lattice has to migrate outward to the surface, leaving cation vacancies;”

Page 14, Line 4-6: “This enzymatic-like reaction-triggered oxidation process of Fe₃O₄ NPs is thought to be analogous to the conventional low-temperature air oxidation of magnetite, in which iron ion migration is probably a rate-limiting step.”

Page 21, Line 8-15: “In summary, the catalytic mechanism of the POD-like activity of Fe₃O₄ nanozymes is elucidated by characterizing the chemical composition and catalytic activity of the Fe₃O₄ NPs recycled from the long-term POD-like catalysis. These studies indicate that not only the surface Fe²⁺, but also the internal Fe²⁺ contribute to the POD-like activity of Fe₃O₄ nanozymes. The Fe²⁺ inside the particle can transfer electrons to the surface, regenerating the surface Fe²⁺ that is constantly involved in the sustained catalytic reaction. This process is usually accompanied by the outward migration of excess oxidized Fe³⁺ from the interior of the crystal, which is considered as a rate-limiting step.”

However, in the introduction, the authors present the peroxidase-like activity of the Fe₃O₄ nanoparticles as unquestionable. However, there are discrepancies, with interesting alternating analyses. For clarity, these alternative points should be presented in the introduction with corresponding references (see: Michel Nguyen, Anne Robert, Bernard Meunier. "Nanozymes" without catalytic activity should be renamed. (2021) hal-03290323).

Response. We are grateful for the suggestion. According to the reviewer’s suggestion, we have added the alternating analyses about the peroxidase-like activity of Fe₃O₄ nanoparticles in the Introduction and marked in red in revised manuscript.

Page 3, Line 18-20: “Although the questioning that the POD-like activity of magnetite is mediated by adventitious metal traces, it has been experimentally verified (including this paper) to exclude the interference of metal traces or the leaching effect of Fe ions in reaction solution.^{10,25-28}”

The methodology presented to addresses the main question of the work is interesting. Firstly, the use of the physicochemical analysis of the recycled of the nanoparticles after “POD”-activity is apparently solid. In a general way, the article is very interesting and has a strong potential to be published. However, in my opinion, several points should be clarified

previously.

Response. Thank you sincerely for your encouraging comments.

Comment 1. Firstly, the “POD”-activity was performed in a really acid pH (3.6). A question rapidly arises: Is a Fe leaching process taking place at this pH? It may be a critical issue due to the estimation and the mechanism acting in the oxidation reactions observed. I suggest to the author using magnetic separation (both more interesting ferrites present strong magnetic response) in the solution of acid buffer to analyze the presence of Fe ions.

Response 1. Thank you sincerely for your insightful question. We studied the Fe ion leaching effect of iron oxide (Fe_3O_4 and $\gamma\text{-Fe}_2\text{O}_3$) nanoparticles (IONPs) in acid buffer (pH=3.6) to assess its influence on the peroxidase-like activity of IONPs. The specific experimental method was described below, and the corresponding analysis of the results (highlighted in red) was added to the revised manuscript and Supplementary Information.

- Experimental Method:

Incubate 5 mg of Fe_3O_4 NPs or $\gamma\text{-Fe}_2\text{O}_3$ NPs in 20 mL of acetate buffer (pH 3.6) for 0-5 days. On days 0, 1, 3 and 5 of incubation, the NPs were recovered by magnetic separation. The supernatant (leaching solution) was transferred to 10 kD ultrafiltration tube and centrifuged at 4500 rcf for 15 min to remove the possible residual NPs. The Fe-ions concentration in the leaching solution was measured using ICP-MS. In addition, the POD-like activity of the recovered NPs and leaching solution after different incubation times were evaluated respectively.

- Results Analysis:

Page 8, Line 16-17: “The impact of leached Fe ions in acidic medium on the catalytic activity of IONPs has been excluded (Supplementary Fig. 8 and Supplementary Table 1).”

Supplementary Information, Page 10, Line 5-16: “In acidic medium, Fe ions may release from the IONPs. Thus it is important to exclude the possible influence of Fe ions leaching effect on the catalytic activity of the recycled IONPs after participating in cyclic catalysis. To test this, we incubated the Fe_3O_4 NPs or $\gamma\text{-Fe}_2\text{O}_3$ NPs in acetate buffer (pH 3.6) and compared the POD-like activity of the leaching solution with that of recovered NPs on days 0, 1, 3 and 5. As shown in Supplementary Fig. 8, the leaching solution within 5 days of incubation showed marginal catalytic activity. Besides, the POD-like activity of the recovered NPs did not show a significantly negative correlation with the incubation time. We also measured the Fe content in the leaching solution using ICP-MS (Supplementary Table 1), which is 1-2 orders of magnitude less than the concentration needed for the Fenton reaction. These indicate that the leached Fe ions does not contribute significantly to the reduction of the catalytic ability of Fe_3O_4 NPs after cyclic catalysis.”

Supplementary Fig. 8 The POD-like activity assessment of the recovered NPs and leaching solution after different incubation times in acidic buffer solution (pH=3.6). (a and b) Fe_3O_4 NPs; (c and d) $\gamma\text{-Fe}_2\text{O}_3$ NPs. Error bars represent standard deviation from three independent measurements.

Supplementary Table 1. Leaching Fe ions concentration of Fe_3O_4 NPs or $\gamma\text{-Fe}_2\text{O}_3$ NPs after incubation in acidic buffer solution (pH=3.6)

Incubation time in acidic buffer solution (days)	Leaching Fe ions concentration ($\mu\text{g}/\text{mL}$)	
	Fe_3O_4 NPs	$\gamma\text{-Fe}_2\text{O}_3$ NPs
0	0.007 \pm 0.000	
1	0.117 \pm 0.026	0.027 \pm 0.002
3	0.224 \pm 0.006	0.029 \pm 0.002
5	0.314 \pm 0.008	0.028 \pm 0.000

Note: The results are expressed as the mean \pm standard deviation of three parallel experiments.

Comment 2. In addition, the Fe^{3+} migration to the internal zones of the nanoparticles and the electron transference to the surface are concluded by comparing the Fe_3O_4 oxidation results and the comparison with the “witness” LiFePO_4 results. However, In my opinion, more direct evidence should be presented. XPS and RAMAN give a global view of the particle oxidation.

A suggest to use an EELS mapping along the nanoparticles exposed to POD-activity after different times in a TEM, taking advantage form the spectral differences of Fe²⁺/Fe³⁺.

Response 2. We are grateful for the suggestion. The electron energy-loss spectra (EELS) is a useful tool for revealing the chemical and oxidation state information of iron oxide at high spatial resolution. According to the reviewer’s suggestion, we measured the EELS of Fe₃O₄ NPs before and after participating in cyclic POD-like catalysis. The measurement result and analysis were presented below, and added to the revised manuscript with red highlights.

Page 9, Line 3-14: “The electron energy-loss spectra (EELS) is a useful tool for revealing the chemical and oxidation state information of iron oxide at high spatial resolution. In general, the peaks of the transition metal *L*-edge shift toward higher energy loss with increasing oxidation state.³⁸ For iron oxide species, the area ratios of Fe *L*₃/*L*₂ also increases with increasing Fe valence.³⁹ As shown in Fig. 1e, both of the Fe₃O₄ NPs before and after five days of cyclic POD-like reactions showed two peaks related to the Fe *L*₃ and *L*₂, with an energy gap of about 13 eV between the two white lines. However, approximately 1.4 eV chemical shift toward high energy loss could be observed from Fe₃O₄ NPs to recycled Fe₃O₄ NPs. In addition, the Fe *L*₃/*L*₂ area ratios also increased from 4.7 to 6.1, which indicates the increase in the Fe oxidation state of Fe₃O₄ NPs after five days of POD-like catalysis.⁴⁰ The similar finding was obtained from the XPS analysis of the recycled Fe₃O₄ NPs on days 0, 1, 3, and 5 of the cyclic catalysis.”

Fig. 1 (e) Comparison of Fe *L*_{2,3} spectra of Fe₃O₄ NPs before and after five days of cyclic POD-like reactions.

Comment 3. In my opinion, a more detailed analysis on the “electron transference” should be presented. Is the electron hope in B site, or the intervalence, acting here? Is the half-metallic nature of magnetite arising from it a key point here? If it is correct, how the migration of Fe³⁺ and the gradual oxidation to maghemite, an isolator, affect this mechanism, since the intervalence is an exclusive feature of magnetite within the iron-oxides family, being depleted with the oxidation.

Response 3. We are grateful for this professional suggestion. The magnetite (Fe_3O_4), as is known, has an inverse spinel structure, with Fe^{3+} occupying tetrahedral (A) sites and equal amounts of Fe^{2+} and Fe^{3+} occupying octahedral (B) sites, written as $(\text{Fe}^{3+})_A[\text{Fe}^{2+}\text{Fe}^{3+}]_B\text{O}_4$. The rapid electron hopping between Fe^{2+} and Fe^{3+} on the B-sites, creating an intermediate valence state of “ $\text{Fe}^{2.5+}$ ”, contributes to the conductivity of magnetite at room temperature, exhibiting a half-metallic nature. Besides, this thermally activated electron delocalization can also result in intervalence charge transfer (IVCT) bands in the visible and near-IR region for magnetite, as shown in Figure 2a and 2b in the revised manuscript. Notably, this electron-hopping process has been reported to be limited to available Fe^{2+} - Fe^{3+} pairs and thus highly dependent on the degree of non-stoichiometry of magnetite.

Oxidizing Fe_3O_4 to non-stoichiometry magnetite ($\text{Fe}_{3-\delta}\text{O}_4$), the Fe^{2+} in B-sites is replaced by vacancies and Fe^{3+} , which can be written as $(\text{Fe}^{3+})_A[\text{Fe}_{2-6\delta}^{2.5+}]_B[\text{Fe}_{5\delta}^{3+}\square_\delta]_B\text{O}_4$ ($0 < \delta < 1/3$). Thus, the number of available Fe^{2+} - Fe^{3+} pairs decreases while isolated Fe^{3+} increases. In addition, the cation vacancies due to oxidation-induced surface migration of excess Fe^{3+} can also disrupt the fast electron-hopping between Fe ions in B-sites. According to the local charge compensation model, each vacancy is electrically equivalent to an extra $-5/2$ charge at one B-site, which has to be neutralized by the excess positive charge at the adjacent B-sites. Thus, each vacancy traps 5 Fe^{3+} and no longer involves in the conduction process. Maghemite ($\gamma\text{-Fe}_2\text{O}_3$) has the highest oxidized spinel structure, where all Fe^{2+} is oxidized Fe^{3+} . To maintain charge neutrality, $1/3$ of the oxidized Fe^{3+} on B-sites must migrate to the surface, leaving cation vacancies, written as $(\text{Fe}^{3+})_A[\text{Fe}_{2/3}^{3+}\text{Fe}^{3+}\square_{1/3}]_B\text{O}_4$.

Therefore, it is concluded that the electron-hopping process can be disturbed due to the reduction of available Fe^{2+} - Fe^{3+} pairs and the formation of cation vacancies when Fe_3O_4 is oxidized to $\gamma\text{-Fe}_2\text{O}_3$. This disturbed hopping process is thought to impair the electron transfer to the surface when Fe_3O_4 nanozymes participate in the sustained POD-like reaction, leading to their depletable catalytic activity.

As suggested by the reviewers, we have added a detailed analysis regarding "electron transfer" to the manuscript in order to make the catalytic mechanism clearer. The following are the details.

Page 14, Line 7-Page 15, Line 3: “As is known, magnetite has an inverse spinel structure, with Fe^{3+} occupying tetrahedral (A) sites and equal amounts of Fe^{2+} and Fe^{3+} occupying octahedral (B) sites, written as $(\text{Fe}^{3+})_A[\text{Fe}^{2+}\text{Fe}^{3+}]_B\text{O}_4$. The rapid electron hopping between Fe^{2+} and Fe^{3+} on the B-sites, creating an intermediate valence state of “ $\text{Fe}^{2.5+}$ ”, contributes to the conductivity of magnetite at room temperature, exhibiting a half-metallic nature.⁵² This electron-hopping process was reported to be limited to available Fe^{2+} - Fe^{3+} pairs and thus highly depends on the degree of non-stoichiometry of magnetite.⁵³ Oxidizing Fe_3O_4 to non-stoichiometry magnetite ($\text{Fe}_{3-\delta}\text{O}_4$) or to $\gamma\text{-Fe}_2\text{O}_3$, the Fe^{2+} in B-sites can be replaced by Fe^{3+} and vacancies, which can be written as $(\text{Fe}^{3+})_A[\text{Fe}_{2-6\delta}^{2.5+}]_B[\text{Fe}_{5\delta}^{3+}\square_\delta]_B\text{O}_4$ (“ \square ” indicates vacancy; “ δ ” indicates vacancy parameter, $0 < \delta \leq 1/3$). Thus, the number of available Fe^{2+} - Fe^{3+} pairs decreases while isolated Fe^{3+} increases. Besides, the formation of cation vacancies due to the surface migration of excess Fe^{3+} can also disrupt the fast

electron-hopping between Fe ions in B-sites. According to the local charge compensation model,⁵⁴ each vacancy is electrically equivalent to an extra $-5/2$ charge at one B-site, which has to be neutralized by the excess positive charge at the adjacent B-sites. Thus, each vacancy traps 5 Fe^{3+} and no longer involves in the conduction process.⁵³ In general, this disturbed electron hopping process caused by the reduction of Fe^{2+} - Fe^{3+} pairs and the formation of cation vacancies is thought to impair the electron transfer to the surface when Fe_3O_4 nanozymes participate in the sustained POD-like reaction, leading to their depletable catalytic activity.”

Comment 4. Finally, why was the electrochemical characterization performed only in the LiFePO_4 system and not in the Fe_3O_4 ? Li also presents a complex chemistry, forming oxidizing radicals with oxygen atoms, which is use for isotopic separation, for example. In order to be use as a “witness” system, I expect a comparative study of the electrochemical response of the Fe_3O_4 nanoparticles.

Response 4. Thank you sincerely for this helpful suggestion. According to the reviewer’s suggestion, the cyclic voltammetric curves of Fe_3O_4 NPs before and after 5 days of cyclic POD-like catalysis were measured and compared with that of LiFePO_4 NPs. The results were analyzed as below and added into the revised manuscript and Supplementary Information, marked in red color.

Page 18, Line 10-Page 19, Line 6: “As a LIBs cathode material, the reversible lithiation and delithiation characteristic of LiFePO_4 contribute to its superior electrochemical performance. We measured the cyclic voltammograms (CV) of the as-prepared LiFePO_4 NPs before and after participating in the cyclic POD-like reactions under different scanning rates in the electrolyte containing Li^+ or Na^+ . As shown in Supplementary Fig. 21, the increase in redox peak currents (I_p) was proportional to the square root of scan rate ($v^{1/2}$), implying a diffusion-controlled process of Li^+ or Na^+ extraction and insertion.⁵⁶ Noticeably, the I_p of the recycled LiFePO_4 NPs (i.e. FePO_4) was obviously reduced compared to LiFePO_4 NPs (Fig. 5d), especially in the electrolyte containing Na^+ (Supplementary Fig. 21 e-h), indicating that the presence of mobile Li^+ contributes significantly to the electrochemical activity. As a comparison with LiFePO_4 NPs, the CV curves of Fe_3O_4 NPs were also measured under the same conditions. As shown in Supplementary Fig. 22, the I_p also exhibits a linear relation with the $v^{1/2}$. However, unlike LiFePO_4 NPs, the I_p of the recycled Fe_3O_4 NPs (i.e. $\gamma\text{-Fe}_2\text{O}_3$) only showed a marginal decrease compared to their counterparts before participating in the catalytic reaction, both of which were found to be similar to the I_p of recycled LiFePO_4 NPs under the same scanning rate (Supplementary Fig. 22 and Fig. 5d). This is probably explained by the lack of freely diffusing ions in the lattice of iron oxide and FePO_4 , which weakens the electron transfer processes in their redox reactions. By contrast, LiFePO_4 NPs exhibited the highest I_p , ascribed to the availability of Li ions in their crystals.”

Supplementary Fig. 21 Cyclic voltammograms of (a and e) LiFePO_4 NPs and (c and g) recycled LiFePO_4 NPs at various scan rates of 0.4~2.0 mV s^{-1} . Plot of peak currents (i_{pa} and i_{pc}) vs. square root of scan rates ($v^{1/2}$) of (b and f) LiFePO_4 NPs and (d and h) recycled LiFePO_4 NPs. The electrolytes used for the experiments in Fig. (a-d) and Fig. (e-f) are lithium acetate buffers and sodium acetate buffer, respectively.

Supplementary Fig. 22 Cyclic voltammograms of (a and e) Fe_3O_4 NPs and (c and g) recycled Fe_3O_4 NPs at various scan rates of 1~20 mV s^{-1} . Plot of peak currents (i_{pa} and i_{pc}) vs. square root of scan rates ($v^{1/2}$) of (b and f) Fe_3O_4 NPs and (d and h) recycled Fe_3O_4 NPs. The electrolytes used for the experiments in Fig. (a-d) and Fig. (e-f) are lithium acetate buffers and sodium acetate buffer, respectively.

Fig. 5 (d) The CV curves of LiFePO₄ NPs and recycled LiFePO₄ NPs in lithium acetate buffers solution, compared with that of the Fe₃O₄ NPs and recycled Fe₃O₄ NPs. Scanning rate is 2 mV s⁻¹.

Comment 5. In addition, small observations arise from the methodology. The nanoparticles prepared by thermal decomposition are presented as “naked-ones”, however, a phase transfer is performed (OA => TMAOH). Are the particles “naked”? More details about this should be given in the Materials and Methods.

Response 5. We are grateful for the suggestion. Tetramethylammonium hydroxide (TMAOH) is used as a phase transfer small molecule agent which can form an electrostatic double layer on the particle surface and stabilize the particles in the aqueous phase. That is, the redispersion of TD-Fe₃O₄@OA NPs in water requires the removal of the surfactant OA layer and replacement with the negative hydroxide ions, which in turn is surrounded by the positive tetramethylammonium counterions. These low polarizing cations favor the stability of Fe₃O₄ NPs in alkaline medium. We initially regarded these ionic double layer encapsulated particles as “naked” particles, which, as the reviewer pointed out, seems not very appropriate. We are very grateful to the reviewer for pointing out this error. According to your suggestion, we have given the relevant details in Materials and Methods and have modified the manuscript accordingly. These revised sentences are listed below and are highlighted in red in the revised manuscript.

Page 6, Line 3-4: Delete the sentence of “To avoid affecting the enzymatic-like activity, all particles were free of the surface coating.”

Page 11, Line 18: “Both Fe₃O₄ NPs have a similar average particle size (~10 nm) with N(CH₃)₄⁺ as surface stabilizer.”

Page 24, Line 13-21: “Synthesis of Fe₃O₄ NPs by thermal decomposition method. Firstly, the OA-coated Fe₃O₄ NPs were synthesized according to the thermal decomposition method we previously reported (denoted as TD-Fe₃O₄@OA NPs).³⁹ The chloroform solvent

was removed from 4 mL of TD-Fe₃O₄@OA NPs (1.627 mg Fe/mL) by rotary evaporation. Then, the OA layer on the particle surface was replaced with 15 mL of 0.5% TMAOH under ultrasonic stirring.⁴⁹ TMAOH is used as a phase transfer small molecule agent which can form an electrostatic double layer on the particle surface and stabilize the particles in the aqueous phase. Next, the reaction system was transferred to 60 mL of the separatory funnel, and 45 mL of chloroform was added. The mixture was thoroughly mixed. After standing for 15 min, the underlying oil phase liquid was discarded. This extraction procedure was repeated three times to fully remove the free OA that was replaced from the particle surface by TMAOH. Finally, the obtained Fe₃O₄ NPs aqueous solution (denoted as TD-Fe₃O₄ NPs) was filtered through a 0.22 μm filter and stored at 4 °C for subsequent experiments.”

Comment 6. Another small point concerns the EPR measurements at very low reaction times. Firstly, when do the authors consider the reaction start point? Second, how did the authors to measure the ESR after 1 min, and even 5 min., taking into account that is necessary to center the sample in the cavity, to measure in the critical coupled condition with the cavity (or matched) and to perform the field scanning (was used only one scan?)

Response 6. Thank you very much for the comments. The EPR experiment was performed as follows: firstly, 10 μg Fe/mL Fe₃O₄ or LiFePO₄ nanoparticles and 30 mM DMPO were added to 300 μL of acetate buffer (pH = 3.6) and mixed well. Then, 0.5% H₂O₂ was added into the above solution and mixed well, which is considered as the reaction start point. Next, the reaction solution was quickly siphoned with a capillary tube, placed in a sample tube and loaded into the resonance cavity. And then, the experimental parameters were adjusted. These two operations took about 45 seconds. The data was scanned and collected once, which took about 12 seconds. Therefore, the whole process was approximately 1 min. Samples for 5 min and 10 min data collection were not prepared additionally, instead, we waited for the sample in the instrument to react to the corresponding time and then collected ESR signals.

In addition, considering that the ESR measurements we conducted were to detect ·OH generation in a relatively short period of time, we further used terephthalic acid (TA) as a fluorescent probe to track the presence of ·OH during the reaction over a longer time period. The detailed experimental steps have been added in Materials and Methods. The new data has been added as Supplementary Fig. 4 and Supplementary Fig. 19 in the revised Supplementary Information. We also have added a few sentences in the revised manuscript to make the statement clear. These revisions were listed below and highlighted in red in the revised manuscript.

Page 27, Line 17-Page 28, Line 4: “**Detection of ·OH using terephthalic acid (TA) as a fluorescent probe.** TA is highly selective and can capture ·OH generated in situ, generating 2-hydroxyterephthalic acid (TAOH) with unique fluorescence around 534 nm. In this work, 0.2 M H₂O₂ was added to 0.2 M acetate buffer (pH = 3.6) containing nanozymes (different materials and concentrations) and 0.5 mM TA. After mixing, the fluorescence signal was

detected on a fluorescence spectrophotometer at specific times in the range of 350 nm-600 nm with an excitation wavelength at 315 nm.”

Page 6, Line 10-12: “As previously reported,^{10,24} the higher catalytic ability of Fe₃O₄ NPs is attributed to the ·OH arising from the surface Fe²⁺-initiated Fenton-like reaction (Supplementary Fig. 3 and Supplementary Fig. 4).”

Page 16, Line 5-7: “The ·OH was shown to be generated in a time- and concentration-dependent manner (Fig. 4d and Supplementary Fig. 19), which is similar to Fe₃O₄ NPs.”

Supplementary Fig. 4 (a) Reaction between terephthalic acid (TA) and ·OH generated by IONPs in the presence and absence of H₂O₂ in 0.2 M acetate buffer (pH = 3.6) after 12 h. (b) Plot of the fluorescence intensity of 2-hydroxyterephthalic acid (TAOH) at 435 nm with reaction time catalyzed by different IONPs. The concentrations of TA, IONPs, and H₂O₂ were 0.5 mM, 4.8 μg/mL, and 0.2 M, respectively. Error bars represent standard deviation from three independent measurements.

Supplementary Fig. 19 (a) Reaction between terephthalic acid (TA) and ·OH generated by LiFePO₄ NPs in the presence and absence of H₂O₂ in 0.2 M acetate buffer (pH = 3.6) after 5 h. The concentrations of TA, LiFePO₄ NPs, and H₂O₂ were 0.5 mM, 24 μg/mL, and 0.2 M, respectively. (b) Plot of the fluorescence intensity of 2-hydroxyterephthalic acid (TAOH) at 435 nm against the reaction time and concentration of LiFePO₄ NPs. Error bars represent standard deviation from three

independent measurements.

In my opinion, the article has potential to be published, and the subject and the approach are very interesting. However, I think that the points listed above should be addressed, especially the first two.

Response. Thank you again for your positive comments and constructive suggestions on our work. We have revised our manuscript carefully by following the guidance you provided. We hope that the revision is acceptable, and your favorable consideration of our manuscript is greatly appreciated.

Reviewer #2 (Remarks to the Author):

The author detailed the mechanism of the POD-like activity of Fe₃O₄ nanozymes and demonstrate that all Fe²⁺ in Fe₃O₄ nanozymes contribute to their POD-like activity. The Fe²⁺ inside the particle transfers electrons to the surface, regenerating the surface Fe²⁺ that is directly involved in the sustained catalytic reaction. An interesting work, however still few question remain to establish this mechanism.

Response. Thank you very much for your positive comments.

Comment 1. Though authors showed the transformation of Fe²⁺ to Fe³⁺ with different time duration and XPS characterization, however still the mechanism for contributing inside Fe²⁺ in their POD like activity.

Response 1. Thank you for your careful review. We apologize for not describing clearly the contribution of inside Fe²⁺ to the POD-like activity of Fe₃O₄ nanozymes. We would like to clarify this with the two additional figures below.

In brief, we propose that Fe²⁺ inside the particle can transfer its own electron to the surface and restore the catalytic activity of surface Fe atoms. Concretely, if, as previously believed, only the active metal atoms on the surface of the nanozymes contribute to the enzyme-like catalytic activity, it is reasonable to assume that when the surface active sites (Fe²⁺) of the Fe₃O₄ nanozymes are depleted (Equation 1) and not capable of timely self-recovery (because the reaction rate constant of Equation 2 is very low), the Fe₃O₄ nanozymes will rapidly deactivate (Supplementary Fig. 9). However, our experimental results revealed that Fe₃O₄ nanozymes can catalyze the POD-like reaction continuously for up to 5 days. To investigate why Fe₃O₄ NPs have such prolonged catalytic ability, we characterized the physicochemical properties of the recovered NPs after cyclic POD-like catalysis. It was found that not only the surface Fe²⁺, but also the interior Fe²⁺ of the Fe₃O₄ nanozymes were gradually oxidized with prolonging the reaction time. Simultaneously, the catalytic activity of the recovered NPs gradually decreases with the increase of their oxidation state. Therefore, we suggest that the involvement of Fe²⁺ inside the particles is responsible for the prolonged catalytic capacity of Fe₃O₄ nanozymes.

Specifically, as shown in Supplementary Fig. 11, when the surface Fe²⁺ is oxidized to Fe³⁺ by the Fenton-like reaction, the adjacent Fe²⁺ inside the particle will continuously transfer its electron outward via the Fe²⁺-O-Fe³⁺ chain in the lattice in order to maintain the catalytic activity of the surface Fe atoms. However, this replenishment of electrons is not infinite. When all of the interior Fe²⁺ are oxidized to Fe³⁺, no electrons can be transferred to the surface of NPs, resulting in the reduction of catalytic activity or even inactivation. We

consider that the oxidation process of Fe_3O_4 due to the enzyme-like catalysis is similar to the conventional low-temperature oxidation of magnetite, accompanied by the surface migration of a small amount of excess Fe^{3+} in the lattice. And the migration rate of Fe^{3+} is thought to depend on the degree of lattice defects in magnetite. Likewise, as described in our work, the migration rate of Fe^{3+} also appears to be a rate-limiting step in the POD-like catalytic reaction of Fe_3O_4 nanozymes. Furthermore, this catalytic mechanism based on electron transfer and ion migration was well confirmed on another model material, LiFePO_4 , in this manuscript.

Supplementary Fig. 9 Only the surface active sites (Fe^{2+}) of Fe_3O_4 nanozymes involve in the POD-like catalytic reaction.

Supplementary Fig. 11 The inside Fe^{2+} transfer electron to particle surface to maintain the POD-like catalytic capacity of Fe_3O_4 nanozymes.

To make the proposed catalytic mechanism more explicit, the above two figures were added in the Supplementary Information as Supplementary Fig. 9 and Supplementary Fig. 11. Also, we have added some explanations in the revised manuscript to make it clear and easier to be understood. These added sentences are listed below and are highlighted in red in the revised manuscript.

Page 10, Line 8-19: “Based on the above characterization results, we conclude that not only the surface Fe^{2+} but also the interior Fe^{2+} of the Fe_3O_4 nanozymes were gradually oxidized by prolonging the reaction time. Simultaneously, the catalytic activity of the recovered NPs gradually decreases with the increase of their oxidation state. Therefore, we suggest that the involvement of Fe^{2+} inside the particles is responsible for the prolonged catalytic capacity of Fe_3O_4 nanozymes. Specifically, as shown in Supplementary Fig. 11, when the surface Fe^{2+} is oxidized to Fe^{3+} by the Fenton-like reaction, the adjacent Fe^{2+} inside the particle will continuously transfer its electron outward via the $\text{Fe}^{2+}\text{-O-Fe}^{3+}$ chain in the lattice to maintain the catalytic activity of the surface Fe atoms. However, this replenishment of electrons is not infinite. When all the interior Fe^{2+} is oxidized to Fe^{3+} , the Fe_3O_4 phase is transformed to $\gamma\text{-Fe}_2\text{O}_3$ without electrons being transferred to the surface, resulting in the reduction of catalytic activity or even inactivation.”

Comment 2. Leaching effect still exists, in that case, the nanozyme can't be re-use in real-life application. This thing should be considered.

Response 2. We appreciate the reviewer's suggestion. This work proposes that Fe_3O_4 nanozymes will undergo a slow oxidation process when participating in the cyclic POD-like reaction, accompanied by the outward migration of excess oxidized Fe^{3+} from the lattice. The reviewer's comment is a thoughtful reminder that we need to consider whether ion leaching effect occur during the reaction and its effect on the catalytic activity of nanozymes. In our reaction system, the leaching effect can be evaluated from two aspects: 1) the Fe ion leaching from IONPs in acidic buffer solution; 2) the surface migration of small amounts of Fe^{3+} accompanying the Fe_3O_4 oxidation process due to the POD-like catalysis.

As for the first aspect, please refer to our **Response to Comment 1 of Reviewer #1**.

To clarify the second point, we measured the Fe ions concentration in the supernatant of the reaction solutions after different cyclic catalytic times. As shown in the **Table below**, the Fe ions concentration in the supernatant of the reaction system after the cyclic catalytic experiments was not significantly different from that of Fe ions leached from Fe_3O_4 after incubation in buffer solution alone, which was one order of magnitude lower than that required for the Fenton reaction.

Therefore, we think that the leaching effect due to catalytic oxidation of Fe_3O_4 nanozymes might be minimal. These minority outward moving Fe ions presumably

coordinate with the surface adsorbed O^{2-} which is ionized by the electrons generated by the oxidation of Fe^{2+} to Fe^{3+} , and form a thin layer of solid solution of $\gamma-Fe_2O_3$ in Fe_3O_4 .

Additional Table for Reviewer: Leaching Fe ions concentration of Fe_3O_4 NPs after different cyclic POD-like catalytic times

Cyclic POD-like catalytic time of Fe_3O_4 NPs (days)	Leaching Fe ions concentration ($\mu g/mL$)
0	0.007 ± 0.000
1	0.114 ± 0.006
3	0.259 ± 0.003
5	0.178 ± 0.027

Note: The results are expressed as the mean \pm standard deviation of three parallel experiments.

Comment 3. Phase dependent activity like why α and γ iron oxide proving different POD-like activity is also missing.

Response 3. We appreciate the reviewer's suggestion. As proposed by reviewer, we have added the explanations of "why Fe_2O_3 in the α and γ phase exhibit different POD-like activity" in revised manuscript and Supplementary Information, which are listed below and highlighted in red.

Page 6, Line 12-18: "The negligible a_{nano} of $\alpha-Fe_2O_3$ NPs compared with $\gamma-Fe_2O_3$ NPs is ascribed to the structural effect of the crystal phases.³⁷ Briefly, $\gamma-Fe_2O_3$ possess cation vacancies at its octahedral positions and most of these vacancies are located on the particle surface, which can favor the adsorption of the substrate H_2O_2 , resulting in a relatively higher POD-like activity (Supplementary Fig. 5). However, these vacancies are not existing on the surface of $\alpha-Fe_2O_3$ due to the change of crystal structure caused by the higher calcination temperature.³⁷"

Supplementary Information, Page 7, Line 3-14: "As shown in Supplementary Fig. 5a, $\gamma-Fe_2O_3$ has a cubic crystal structure of inverse spinel type with cation vacancies in octahedral positions. Most of these cation vacancies are located on the particle surface.⁶ As a result, the anionic sites of O on the surface are exposed, allowing substrate H_2O_2 to be easily adsorbed. The adsorbed H_2O_2 undergoes acid-like dissociative chemisorption, producing H^+ and OOH^- . Then, OOH^- reacts with Fe^{3+} to form $FeOOH^{2+}$ (Eq. S1 in Supplementary Fig. 5b). Subsequently, the OOH^- in $FeOOH^{2+}$ donates an electron to Fe^{3+} to form Fe^{2+} (Eq. S2 in Supplementary Fig. 5b), which triggers the Fenton-like reaction (Eq. S3 in Supplementary Fig. 5b), resulting in a relatively higher POD-like activity of $\gamma-Fe_2O_3$ NPs (Eq. S4 in Supplementary Fig. 5b). However, these cation vacancies are not existing on the surface of

α -Fe₂O₃⁶ due to the change of the inverse spinel structure caused by the high calcination temperature (650 °C), leading to a lower affinity to H₂O₂ and the negligible POD-like activity of α -Fe₂O₃.”

Supplementary Fig. 5 (a) Diagram of the crystal structure of γ -Fe₂O₃ and α -Fe₂O₃. (b) Reaction equations for the POD-like activity of IONPs. (The Figures are cited from Ref. ⁶)

Comment 4. Few recently published articles (e.g. <https://doi.org/10.1002/adma.202107088>; <https://doi.org/10.1002/ange.202112453>; <https://doi.org/10.1039/C9TB00989B>) should be discussed and cited.

Response 4. Thank you for your careful review. According to the reviewer’s suggestion, some recently published articles have been discussed and cited in the revised manuscript. And these cited articles have been added to the Reference List and were highlighted in red.

Page 3, Line 6-13: “For example, single-atom nanozymes centered on different metals species have been synthesized with well-defined structures and coordination environments, which facilitate the identification of catalytic centers and unravel the catalytic mechanisms at the atomic level.¹⁶⁻¹⁹ Besides, the high substrate selectivity of nanozymes has been achieved by the bionic principle of natural substrate channeling and stepwise screening or by the molecular blotting techniques.^{20,21} Given the intricate structure-activity relationships and restricted characterization techniques, however, it is still challenging to understand the explicit mechanism of most nanozymes.^{3,4}”

Page 6, Line 12-18: “The negligible a_{nano} of α -Fe₂O₃ NPs compared with γ -Fe₂O₃ NPs is ascribed to the structural effect of the crystal phases.³⁷ Briefly, γ -Fe₂O₃ possess cation vacancies at its octahedral positions and most of these vacancies are located on the particle surface, which can favor the adsorption of the substrate H₂O₂, resulting in a relatively higher POD-like activity (Supplementary Fig. 5). However, these vacancies are not existing on the surface of α -Fe₂O₃ due to the change of crystal structure caused by the higher calcination temperature.³⁷”

Reviewer #3 (Remarks to the Author):

The manuscript "The Depletable Peroxidase-like Activity of Fe₃O₄ Nanozymes Accompanied with Phase Transformation Triggered by Separate Migration of Electrons and Iron Ions" by Deng et al. reports on an aspect of the catalytic activity of Fe₃O₄ nanozymes, namely the electron transfer via a Fe(2⁺)-O-Fe(3⁺) chain leading to a reduction of ion charge at the NP surface. This highlights the importance of bulk-surface atom interaction in the catalytic reaction. Overall, the findings are interesting and definitely relevant for the research community focusing on catalytic materials.

Response. Thank you very much for your positive comments.

Comment 1. However, the focus of the work is quite narrowly set on the specific aspect of surface Fe(2⁺) activation and the impact of surface passivation on catalytic activity.

Response 1. Thank you for your careful review. To make this work more clearly understood by reviewers, we have summarized the main ideas and innovations of this work as follows.

• Main ideas of this work:

1. When Fe₃O₄ NPs exert their POD-like activity, not only the surface active Fe²⁺ works, but also the internal Fe²⁺ transfers electron to the surface, driving the continuation of the catalysis.

2. The surface transfer of internal electrons does not proceed indefinitely. Fe₃O₄ will eventually be oxidized to γ-Fe₂O₃, leading to the weakening of POD-like activity.

3. The outward migration of the excess Fe ions in the lattice is associated with the oxidation process of Fe₃O₄ caused by POD-like catalytic reaction. The migration rate has been considered as a rate-limiting step in the catalytic reaction.

4. As a proof-of-concept, this catalytic mechanism based on electron transfer and ion migration has been well confirmed on another model material, LiFePO₄, in this work.

• Innovations of this work:

1. This work reveals that internal atoms may also contribute to nanozyme-catalyzed reactions even though these reactions occur on the surface of NPs.

2. This work points out the self-depleting property of Fe₃O₄ nanozymes, which is different from natural HRP.

3. Our findings may inspire the study of the catalytic mechanism of other kinds of nanozymes, and also provide a theoretical basis for a more rational design, regulation and application of nanozymes in the future.

Comment 2. The authors suggest a possible mechanism that leads to passivation of the nanozyme without providing any alternative for how this could be remediated.

Response 2. We are grateful for the suggestion. As stated by the reviewer, it is important in the field of catalysis to prevent and remediate catalyst surface passivation. And many strategies have been exploited to address this problem. However, nanozymes as a young research field, the intrinsic catalytic mechanisms of most types of nanozymes are still unclear, which hinders their rational design and efficient application in the fields of medicine, chemistry, food, agriculture and environment. Therefore, the focus of this work is to reveal the catalytic nature of the POD-like activity of Fe₃O₄ NPs, which is the first discovered and most applied nanozyme.

A deeper understanding of the catalytic mechanism can facilitate the controlled and purposeful design of nanozymes. According to the findings of our work, Fe₃O₄ NPs can be oxidized and suffer from the chemical component changes when they exert the POD-like activity for a long duration, leading to a significant weakening of their catalytic capacity. Therefore, we think that it is possible from the perspective of "how to make the oxidized iron ions in the lattice to be reduced again" to solve this problem.

The heterogeneous Fenton reaction, such as heterogeneous photo-Fenton, may be an effective solution. For example, the synthesis of boron-doped reduced graphene oxide wrapped Fe₃O₄ (Fe₃O₄@B-rGO) composite for the degradation of Bisphenol A (BPA) has been reported (*Applied Surface Science* 544 (2021) 148886). It was shown that the doping of boron atoms could enhance the band gap of rGO, which made B-rGO can be used as semiconductor to generate electrons in the conduction band and h⁺ in the valance band upon visible light illumination. The photogenerated electrons can not only be captured by Fe₃O₄ to accelerate the transformation of Fe(III) to Fe(II) so as to promote the utilization rate of H₂O₂, but can also directly react with O₂ to produce O₂^{•-} (**Additional Figure 1 for Reviewer**). We think that such strategy of coupling with inorganic photocatalysts or organic semiconductors could be adapted to modulate the catalytic activity of some metal oxide nanozymes, such as Fe₃O₄, CeO₂, CuO, MnO₂.

Additional Figure 1 for Reviewer: Simplified degradation mechanism of Bisphenol A in the photo-Fenton system with Fe₃O₄@B-rGO. (<https://doi.org/10.1016/j.apsusc.2020.148886>)

Interestingly, we noticed that the similar strategy described above was implemented in a recently published paper (*Nano-Micro Lett.* (2022) 14:101.) In this paper, the authors integrate plasmonic metal (Gold nanorods) with semiconductor (CeO₂ NPs) nanoenzyme structures (STGC) to achieve photon-driven sub-nanostructural transformations via regulating their electronic properties to initiate the local atomic reconstruction, so as to facilitate the catalytic activity regulation of nanozymes. In detail, as shown in **Additional Figure 2 for Reviewer**, upon NIR irradiation, the interband transition of Au is excited to produce electron-hole pairs, and the hot electrons transfer from excited Au to the conduction band (CB) of CeO₂ to participate in the reduction of Ce⁴⁺ and produce active oxygen vacancies, rebuilding the POD-like activity of STGC for H₂O₂ decomposition.

Additional Figure 2 for Reviewer: Schematic illustration of the design and photon-driven sub-nanostructural transformation of STGC through the direct electron transfer. (<https://doi.org/10.1007/s40820-022-00848-y>)

Comment 3. This focus makes the work very limited with respect to applicability and possible pathways of improving the performance of catalysts.

Response 3. We appreciate the reviewer's suggestion. We would like to emphasize that the focus of our work is on the study of the catalytic mechanism of the POD-like activity of Fe₃O₄ NPs, which is the first discovered and most applied nanozyme. Because only with a clear insight into the catalytic mechanism of nanozyme can targeted improvements be made. Given the workload of one paper and the space limitation of the journal, this manuscript does not cover how to mitigate the self-oxidation of Fe₃O₄ due to enzyme-like catalysis, which, as mentioned by reviewer, is a new subject worth investigating. But we tried to answer this question in **Comment 2** and hope that the response is acceptable. In the follow-up research work, we will focus on how to remedy the self-depleting characteristics of Fe₃O₄ nanozymes, and look forward to publishing the related research results as soon as possible.

Comment 4. Although the electron transfer mechanism is intriguing in itself and the comparison with LiFePO_4 makes a strong point, I see a lack of generality that would make this work worthy of being published in Nature Communications. Therefore, I suggest considering this work for a more specialised journal.

Response 4. We appreciate the reviewer's valuable comments and criticisms. we did our best to answer your constructive questions, and hope that our response above is acceptable.

Minor comments:

1. Figs. 1(e) and (h) are too small. Details a very hard to identify.

Response. We are grateful for the suggestion. Based on the reviewer's suggestion, we have enlarged these two figures, and renumbered them as Fig. 1 (f) and Fig. 1 (i) in the revised manuscript, as shown below.

Fig. 1 The synthesis of IONPs and cyclic POD-like catalysis. (a) Illustration of the synthesis process of IONPs. (b) The specific activity (a_{nano}) of these three IONPs with TMB as colorimetric substrates. (c) Diagram of the cyclic catalysis assay. (d) Kinetic study of a_{nano} values of Fe_3O_4 NPs with the days of cyclic catalytic reaction. **Error bars represent standard deviation from three independent measurements.** (e) Comparison of Fe $L_{2,3}$ spectra of Fe_3O_4 NPs before and after five days of cyclic POD-like reactions. (f) The fitted Fe2p XPS spectra of Fe_3O_4 NPs recycled after catalysis on days 0, 1, 3, and 5. (g) The Fe L-edge NEXAFS spectra of Fe_3O_4 NPs and recycled Fe_3O_4 NPs after 5 days of catalysis in comparison with the reference spectra of FeSO_4 and Fe_2O_3 . (h) Raman spectra of Fe_3O_4 NPs recycled after catalysis on days 0, 1, 3, and 5. (i) TEM, HRTEM images and SAED pattern of Fe_3O_4 NPs and recycled Fe_3O_4 NPs after 5 days of catalysis.

2. Label of y-axis in Fig. 1(b) is " b_{nano} " and should be " a_{nano} ".

Response. We are grateful to the reviewer for the careful checking. After our recheck, in fact, the label of y-axis in Fig. 1(b) should indeed be " b_{nano} ". According to the standardized method for determining the catalytic activity of peroxidase-like nanozymes (Eq. 1 and 2):

$$b_{\text{nano}} = \frac{V \times (\Delta A / \Delta t)}{\varepsilon \times l} \quad (1)$$

$$a_{\text{nano}} = \frac{\Delta b_{\text{nano}}}{\Delta m_{\text{Fe}}} \quad (2)$$

b_{nano} is the POD-like catalytic activity of IONPs expressed in units (U);

a_{nano} is the specific activity expressed in units per milligram nanozymes ($\text{U} \cdot \text{mg}^{-1}$);

m_{Fe} is the total Fe element mass contained in added NPs (mg);

Therefore, in the Fig.1 (b), a_{nano} can be calculated by the slope of the curve plotting the b_{nano} against different masses of NPs ($\frac{\Delta b_{\text{nano}}}{\Delta m_{\text{Fe}}}$).

Additionally, following the reviewer's kind reminder, we also double-checked the label of x- and y-axis of the other Figures in the revised manuscript to ensure their correctness.

REVIEWERS' COMMENTS

Reviewer #1 (Remarks to the Author):

In my opinion, the authors significantly improve the quality of the work, addressing all my comments in the previous revision process. I keep my opinion about the quality of the work, and with all the changes/corrections that improved the scientific quality of the work, I recommend this work for publication.

Reviewer #2 (Remarks to the Author):

The revised manuscript is highly improved.